# Representation of the Tropical Stratospheric Zonal Wind in Global Atmospheric Reanalyses

Y. Kawatani[1], K. Hamilton[2], K. Miyazaki[1], M. Fujiwara[3], J. Anstey[4]

[1]Japan Agency for Marine-Earth Science and Technology, Yokohama, 236-0001, Japan
[2]International Pacific Research Center, University of Hawaii, Honolulu, 96822, USA
[3]Faculty of Environmental Earth Science, Hokkaido University, Sapporo, 060-0810, Japan
[4]Canadian Centre for Climate Modelling and Analysis, University of Victoria, Victoria, V8W 2Y2, Canada

*Correspondence to*: Y. Kawatani (yoskawatani@jamstec.go.jp)

**Abstract.** This paper reports on a project to compare the representation of the monthly-mean zonal wind in the equatorial stratosphere among major global atmospheric reanalysis datasets. The degree of disagreement among the reanalyses is characterized by the standard deviation (SD) of the monthly-mean zonal wind and this depends on latitude, longitude, height and the phase of the quasi-biennial oscillation (QBO). At each height the SD displays a prominent equatorial maximum, indicating the particularly challenging nature of the reanalysis problem in the low-latitude stratosphere. At 50–70 hPa the geographical distributions of SD are closely related to the density of radiosonde observations. The largest SD values are over the central Pacific, where few *in situ* observations are available. At 10–20 hPa the spread among the reanalyses and differences with *in situ* observations both depend significantly on the QBO phase. Notably the easterly-to-westerly phase transitions in all the reanalyses except MERRA are delayed relative to those directly observed at Singapore. In addition, the timing of the easterly-to-westerly phase transitions displays considerable variability among the different reanalyses and this spread is much larger than for the timing of the westerly-to-easterly phase changes. The eddy component in the monthly mean zonal wind near the equator is dominated by zonal wavenumber 1 and 2 quasi-stationary planetary waves propagating from mid-latitudes in the westerly phase of the QBO. There generally is considerable disagreement among the reanalyses in the details of the quasi-stationary waves near the equator. At each level, there is a tendency for the agreement to be best near the longitude of Singapore, suggesting that the Singapore observations act as a strong constraint on all the reanalyses. Our measures of the quality of the reanalysis clearly show systematic improvement over the period considered (1979–2012). The SD among the reanalysis declines significantly over the record, although the geographical pattern of SD remains nearly constant.

## 1. Introduction

The dynamics governing the circulation in the tropical stratosphere have attracted much interest over the years (e.g. Wallace, 1973; Baldwin et al., 2001). As in other regions of the middle atmosphere, the day-to-day and higher-frequency variations of the flow in the tropical stratosphere are believed to be dominated by a spectrum of vertically propagating waves excited in

the troposphere. What makes the low latitude stratosphere so remarkable is that the forcing of the zonal-mean flow by these waves leads to the very large-amplitude, low-frequency quasi-periodic cycle known as the quasi-biennial oscillation (QBO). In the tropical stratosphere, the QBO clearly dominates other aspects of interannual variability and even swamps the annual and semiannual variations in the zonal-mean circulation, at least up to ~3 hPa. Although rooted in the tropics, the QBO has

5 global impacts. The QBO strongly influences interannual variations in circulation and composition throughout the stratosphere. The QBO also affects the circulation at the Earth's surface and is an important consideration in extended-range weather forecasts (e.g. Baldwin et al., 2001). Recently, Yoo and Son (2016) indicate the QBO exerts greater influence on the Madden-Julian Oscillation than does the El Niño-Southern Oscillation.

The state of the QBO up to the middle stratosphere can be characterized by the time series of monthly mean, near-equatorial

zonal winds at levels between 10 and 70 hPa maintained by the Free University of Berlin (FUB, e.g. Naujokat, 1986) since 1953. The monthly values in the FUB series are based on operational balloon soundings, and the FUB record has been stitched together from such observations at Canton Island (2.8°S, 172°W from January 1953 to August 1967), Gan (0.7°S, 73°E from September 1967 to December 1975), and Singapore (1.4°N, 104°E since 1976). The high quality of these balloon data, and the close proximity of the stations to the equator, has led FUB series to be widely used, despite being based on only

a single station each month (and despite modest inhomogeneities that the changes of station location may introduce into the record; see Section 3 below).

Global atmospheric analyses that assimilate all available satellite remote sensing and *in situ* observations are another potential source of information about the QBO and other aspects of the circulation in the tropical stratosphere. Data assimilation is the technique for combining different observational data sets with a model, by considering the characteristics

of each measurement and taking into account errors in both the measurements and the model (e.g., Kalnay, 2003). Advanced data assimilation schemes like the 4D-Var technique use the information provided by various measurements, such as radiosonde and satellite-derived measurements, and propagates it, in time and space, from a limited number of observable variables to a wide range of meteorological variables to provide global fields that are dynamically consistent and in agreement with the observations. Meteorological reanalyses have been conducted at operational centers using various

approaches, which ingest a variety of observations over the period of each reanalysis product. Differences in the forecast model, assimilated measurements, and data assimilation technique used for producing reanalysis datasets can lead to differences in their representation of the mean state, variability, and long-term trend of atmospheric fields.

A number of factors combine to make the global meteorological analysis process particularly challenging in the tropical middle atmosphere.

(i) One challenge is the relative paucity of *in situ* data, even if attention is restricted to levels at and below 10 hPa (i.e. the usual upper bound for most operational balloon soundings). There are large ocean areas in the tropics with no balloon observations. Even over land areas, the observations in the stratosphere at many stations are sparse. At stratospheric levels, the zonal wind measurements at many stations near the equator tend to have short overall records, or records with many

months that have no observations (or not enough to compute a stable monthly mean). This leads to gaps in time series of monthly-mean winds (e.g. Hamilton, 1984; Kawatani and Hamilton, 2013).

(ii) Near the equator the Coriolis parameter is small; so observations of the temperature from satellite remote sensing do not constrain the wind field as strongly as at higher latitudes. Even if we assume the near-equatorial flow really is close to thermal wind-balance (Reed, 1962; Randel et al., 1999) the computed geostrophic wind shears are extremely sensitive to small errors in the observed temperatures.

(iii) The flow in the tropical stratosphere exhibits variations on very small vertical scales. This limits the usefulness of the relatively coarse-resolution satellite remote-sensing temperature retrievals. Even the monthly-mean zonal wind in this region displays thin layers where the wind can change by ~30 m s$^{-1}$ over ~3 km. Satellite radiances used in global assimilations have an effective vertical resolution of several kilometers (e.g. Huesmann and Hitchman, 2003). Huesmann and Hitchman (2003) note that in such shear regions the assimilation scheme will have to reconcile the strong wind-shears measured directly by balloons with the somewhat weaker thermal winds consistent with the satellite derived temperature gradients (artificially damped due to the coarse vertical resolution).

(iv) Modern data assimilation approaches use general circulation model (GCM) simulations to obtain background (i.e. first guess) information and to determine data assimilation analysis increments. The tropical stratosphere is perhaps the only region of the atmosphere where most free-running GCMs have simulations with zero-order errors in the zonal-mean circulation (i.e. nothing even resembling a realistic QBO). This is because most GCMs display fairly steady, weak prevailing zonal winds due to failure to reproduce a spontaneous QBO. Such models typically relax any QBO-like zonal winds in the initial condition towards the model climatology (e.g. Hamilton and Yuan, 1992; Saha et al., 2006; Boer and Hamilton, 2008) and so the persistent model bias could act against the data assimilation analysis increments and damp the QBO signals as introduced by data assimilation in the reanalysis fields. It should be noted that the situation is now slowly changing and several models have simulated a fairly realistic spontaneous the QBO in the equatorial stratosphere. Such models typically either employ fine vertical resolution as well as representation of momentum transports from small-scale non-stationary gravity waves either through high horizontal resolution (Kawatani et al. 2010 and reference therein) or by parameterization (Orr et al. 2010; Kawatani and Hamilton 2013; Anstey et al. 2016).

These special challenges in data assimilation for the tropical stratosphere have led to problems in actual global assimilation products, although there have been clear improvements over the last two decades in the ability of analysis systems to represent this region of the atmosphere. Trenberth (1992) documents a total misrepresentation of the QBO in the tropical stratosphere of ECMWF operational global analyses in the early 1980's, but finds that the situation improved considerably after changes in the assimilation scheme were introduced in May 1986.

In the 1990's some major meteorological centers started producing retrospective reanalysis products that employed multivariate statistical data assimilation methods such as optimal interpolation (OI) and three-dimensional variational (3D-VAR) scheme to combine all available data over some extended period. Such reanalyses have obvious advantages for research applications requiring the most homogeneous possible data set throughout an extended period. Two of the early

major products were the ECMWF "ERA-15" reanalysis (covering 1979 to 1994) and the first NCAR/NCEP reanalysis (covering 1948–near present). The QBO in the tropical stratosphere in these products was examined by Pawson and Fiorino (1998, 1999) and Huesmann and Hitchman (2001, 2003) who found that the reanalyses displayed equatorial QBO variations that were clearly smaller in amplitude than in the real atmosphere. Randel et al. (2004) indicated that QBO variations in

temperature and zonal wind were underestimated in most reanalyses available at that time, as compared to Singapore radiosonde data. They found that only ERA-40 had realistic zonal wind amplitudes above 30 hPa.

Some centers have now produced multiple reanalyses covering the same (or overlapping) periods; these more recent products are derived with updated data assimilation systems. Notable among these are the ECMWF ERA-40 (September 1958 to August 2002) and ERA-Interim (1979 to present). These products have improved representation of the QBO

amplitude over that in ERA-15 (e.g. Pascoe et al., 2005; Baldwin and Gray, 2005). Baldwin and Gray (2005) compared the FUB zonal wind with ERA-40 data, and showed the zonal mean equatorial ERA-40 wind is quite close to that indicated in the FUB data.

Additional major reanalysis datasets have become available recently. The present paper reports on one of the studies contributing to the SPARC Reanalysis Intercomparison Project (S-RIP; Fujiwara et al., 2013; Errera et al., 2015), which are

focused on evaluating reanalysis output for the stratosphere. We compare the representation of the circulation in the tropical stratosphere among several contemporary reanalysis products and validate the reanalyses against *in situ* observations. We have mostly limited our attention to just a single aspect of the circulation, namely the monthly-mean zonal wind. We limit our consideration to low latitudes (20°N–20°S) and a height range (10–70 hPa) in which the QBO is strong and where substantial balloon data are available for comparison. We have also restricted our attention to the period starting in 1979

when NOAA operational satellite radiance observations became available and were incorporated as an important data source in all the reanalyses. We appreciate that other aspects of the reanalyses in this region are also of interest, notably the ability of the reanalyses to represent accurately the large-scale transport circulation in the stratosphere (e.g. Coy and Swinbank, 1997; Abalos et al. 2015; Miyazaki et al. 2015), but these aspects will be left for future research.

Each reanalysis uses a different forecast model and assimilation scheme, and the types and numbers of assimilated

observational data are also different among reanalyses. Furthermore, it is not feasible to determine exactly what observational data were actually assimilated at each data assimilation analysis step (e.g., what data quality control and bias correction procedures were actually applied). These complications make it somewhat difficult to attribute conclusively all the differences among the reanalyses products. However, it is interesting to investigate representations of key phenomena in the reanalyses. We hope such investigation will contribute to basic understanding and to improving future reanalysis

products. We will show that, even with our somewhat narrow focus, some interesting conclusions concerning the representation of the tropical stratospheric circulation will emerge.

The outline of this paper is as follows. Section 2 briefly describes the reanalysis products that we evaluated and the station balloon data we employed. Section 3 investigates differences of the tropical zonal wind among reanalyses. Section 4

discusses evolution of the differences among reanalyses with time as different data sources become available. Section 5 summarizes the study and provides concluding remarks.

## 2. Reanalysis and radiosonde observation data

We analyzed monthly mean zonal wind and temperature in nine sets of global reanalyses data [NCEP-1 (Kalnay et al., 1996),
NCEP-2 (Kanamitsu et al., 2002), NCEP-CFSR (Saha et al., 2010), ERA-40 (Uppala et al., 2005), ERA-I (Dee et al., 2011), JRA-25 (Onogi et al., 2007), JRA-55 (Kobayashi et al., 2015), MERRA (Rienecker et al., 2011) and MERRA-2 (Molod et al., 2015)]. Monthly mean data for 20°S–20°N and 10–70 hPa after 1979 are mainly analyzed, except for MERRA-2 which has data only after January 1980. Data before December 2012 are investigated, except for ERA-40 and NCEP-CFSR which are available until August 2002 and December 2010, respectively. With the exception of MERRA-2, none of the global
dynamical models used in the assimilations would simulate the QBO when in free running mode. The dynamical model used in producing the MERRA-2 reanalyses is able to simulate a spontaneous QBO in the tropical stratosphere because it includes quite strong parameterized momentum fluxes from non-orographic gravity waves (Fig. 3 of Molod et al., 2015).

One key observational data set we employed for comparisons is the near-equatorial monthly mean values of operational balloon-borne radiosonde observations compiled by FUB (http://www.geo.fu-berlin.de/en/met/ag/strat/produkte/qbo/). The
FUB data during 1979–2012 (i.e. period analyzed in this study) are based on Singapore observations.

We have also used monthly-mean values of the zonal wind at many other radiosonde stations provided in the Integrated Global Radiosonde Archive (IGRA: https://www.ncdc.noaa.gov/data-access/weather-balloon/integrated-global-radiosonde-archive, Durre et al., 2006). For 10°S–10°N IGRA includes balloon data from ~220 stations. When twice daily observations are available, the IGRA provides monthly means of observations at 00 and at 12 UTC separately. As in Kawatani and
Hamilton (2013), where possible, the monthly values for 00 and 12 UTC are averaged, but the single values are used when only 00 or only 12 UTC data are available. The diurnal cycle of the wind in the lower stratosphere is expected to be small, and indeed we have found that generally the monthly mean zonal wind measured at 00 UTC data is nearly identical to that at 12 UTC.

In order to compare the data among reanalyses which have different spatial resolutions, all reanalysis variables are
interpolated linearly to the ERA-I grid (i.e. 1.5° resolution in longitude and latitude, 37 vertical levels from 1000 to 1 hPa). When reanalysis data are compared with FUB/IGRA observational data, the reanalysis values are interpolated to the station locations.

## 3. Differences of the tropical zonal wind among reanalyses
### 3.1. Dependence of differences on longitude and height

Figure 1 shows the FUB zonal wind and the zonal wind in each reanalysis included in this study at Singapore (104°E, 1.4°N) from 1979 to 2012. All the reanalyses have a reasonably close resemblance to the FUB observations. Notably each reanalysis

clearly captures the basic features of the QBO seen in the FUB data including the cycle-to-cycle variation in period and amplitude.

Figure 2 displays time variations of the zonal wind in each reanalysis at 10, 20, 30, 50, and 70 hPa over Singapore compared with the FUB observations. Also shown are the root mean square (RMS) differences between FUB and each reanalysis zonal wind averaged from 10 to 70 hPa. Reanalysis zonal winds are generally close to the FUB zonal wind. The RMS differences display somewhat noisy variations but an overall trend to smaller values over time is apparent. The zonal winds in NCEP-1 and NCEP-2 are generally underestimated, especially in the 1980's over 20–50 hPa. The most obvious anomaly in the reanalysis winds at 10 hPa is found in MERRA-2, which exhibits spurious semi-annual variations in the 1980's and in late 1993 particularly during easterly phase of the QBO (Figs. 1g and 2a). The downward propagation of westerly semiannual oscillation (SAO) phases is enhanced during these periods, which could be caused by overly strong gravity wave forcing (Fig. 3 of Molod et al., 2015). Coy et al. (2016) note that MERRA-2 appears to overemphasize the annual cycle before 1995. The RMS differences from FUB values are smallest in ERA-I, while those in NCEP-1 and NCEP-2 are much larger than those in the other reanalyses. MERRA-2 represents large RMS differences in 1980's (Fig. 2f), mainly due to the enhanced SAO at 10 hPa. On the other hand at 30–50 hPa, the MERRA-2 zonal winds show improved representation of the QBO compared to MERRA (Lawrence Coy et al., 2016). The NCEP-CFSR actually uses the ERA-40 stratospheric wind profiles as bogus observations in the tropics from 1 July 1981 to 31 December 1998 to obtain a reasonable QBO (Saha et al., 2010). We confirmed the differences between NCEP-CFSR and ERA-40 during this period are nearly zero. For these reasons, our study focuses mainly on the five reanalyses: ERA-40, ERA-I, JRA-25, JRA-55 and MERRA.

In order to quantify the spread among reanalyses, the standard deviations (SD) among the reanalyses are calculated as follows:

$$SD = \sqrt{\sum_{i}^{N}(u_i - [u])^2 / N} \tag{1}$$

where $i$ labels the individual datasets and there are $N$ datasets included. The square brackets [ ] denote the mean over all $N$ reanalyses. The SD is calculated for each month using monthly mean zonal wind. Because the ERA-40 data are only provided until August 2002, attention will sometimes be restricted to the SD during January 1979 to December 2001.

The latitude-height distributions of the zonal mean and time mean of the SD among five reanalyses (1979–2001) are displayed in Fig. 3a. Large SD is seen in the upper troposphere and the stratosphere at low-latitudes, while the SD in the extratropical regions is much smaller. Figures 3b and 3c depict the SD among four reanalyses (ERA-I, JRA-25, JRA-55, and MERRA) and among the three latest reanalyses (ERA-I, JRA-55, and MERRA) in 1979–2012. These are discussed in the next section. In any case, the distributions of the SD are quite similar.

In order to examine the SD in more detail later, the SD is divided into zonal mean components ($\bar{u}$; overbar denotes the zonal mean) and components of deviation from zonal mean ($u'$: hereafter, referred to as eddy components). Substituting $u_i = \bar{u}_i + u_i'$ and $[u] = [\bar{u}] + [u']$ in Eq. (1) and taking the zonal mean yields:

$$\overline{(SD)^2} = \sum_i^N \overline{(\overline{u}_i - [\overline{u}])^2} / N + \sum_i^N \overline{(u'_i - [u'])^2} / N \qquad (2)$$

Here, the first term on the right side represents the zonal mean variance among reanalyses due to the zonal mean wind component and second term exhibits the zonal mean variance attributable to eddy components.

Figure 4 shows horizontal distributions of the SD at 10, 20, 30, 50, and 70 hPa. At 70–50 hPa, large SD is seen in the Indian Ocean, to the east of the maritime continent, the Atlantic, and especially in the central Pacific. At 10–30 hPa, on the other hand, the SD becomes more zonally uniform with increasing height.

## 3.2. Dependence in the lower stratosphere

Satellite radiance observations are distributed much more uniformly and homogeneously than *in situ* balloon observations, so it might be expected that satellite data assimilation would not have a large effect on the spatial distribution of the SD among reanalyses. We may expect the SD geographical distributions to be related primarily to the availability of station observations. Figure 5a shows the locations of all radiosonde stations in the IGRA (magenta dots indicate the locations of the stations shown later in Fig. 6). We calculated the fraction of months during 1979–2001 with valid monthly mean wind data at each level at each IGRA station. Figures 5b–f display the fractional data coverage at each radiosonde station with at least some useful data from 10 to 70 hPa (the 100% coverage at Singapore in the FUB data is also indicated). The contours in these panels reproduce the values of the SD shown in Fig. 4. Over 10°N–10°S, there are about 220 radiosonde stations (Fig. 5a), but only a fairly small fraction have significant data coverage at the stratospheric levels of interest here, particularly at 10 hPa.

At 70–50 hPa, distributions of the SD appear to be negatively correlated with the density of radiosonde observations. The SD shows the local minima in the zonal direction near the locations with high observational density, such as the vicinity of Sumatra, Borneo and the Malay Peninsula, and over much of South America. Large SD values are found over an extensive region in the eastern and central Pacific where few radiosonde observations are available.

Observational information is propagated with time and space through analysis steps and forecast steps in data assimilation. In analysis steps, the background error covariance matrix is used to determine the spatial structure and the magnitude of analysis increment. Advanced data assimilation techniques such as 4D-VAR (used in JRA-55 and ERA-I) are expected to provide more efficient constraints even for remote points, because of the use of flow-dependent analysis. The different data assimilation methods used in each reanalysis would contribute to the SD even if each reanalysis assimilated the same observational data. In addition, the SD may also reflect differences among the climatologies of the GCMs used in the analysis systems.

Next, we consider the time variations of the spread among reanalyses at individual places and its relation to the availability of local radiosonde data. Each panel in Fig. 6 shows the time variations of monthly mean 70 hPa zonal wind measured at a particular radiosonde station together with the values from each of the five reanalyses and the SD among the reanalyses. The

blue/red colors of the SD curves in each panel denote times with/without radiosonde data at the station. In the upper right corner of each bottom panel, the total data coverage (black number) and the average value of the SD during periods with (blue number) and without (red number) radiosonde data at each station are shown.

At data rich stations with coverage ≥ 80% such as Thiruvananthapuram (Fig. 6c), Singapore (Fig. 6d), Chuuk (Fig. 6f), Majuro (Fig. 6g), and Ascension (Fig. 6m), reanalyses zonal winds generally agree well with the direct observations, and the SD among reanalyses is small. On the other hand, at Christmas Island (Fig. 6h) where no observational data are available from these 23 years, the SD is large throughout the record. There are no other stations with available data near Christmas Island (Fig. 5a), which may also contribute to the extended period of large SD there.

Menado (Fig. 6e) and San Cristobal (Fig. 6i) demonstrate that the reanalysis SD at the locations of near-equatorial stations (within 1.5° of the equator; Fig 5a) can be appreciably reduced during times when radiosonde data from these stations are available. Bogota (Fig. 6j) and Manaus (Fig. 6k) are good examples off the equator also showing smaller SD during months with balloon data than without these data.

At Nairobi (Fig. 6a), an unrealistic spike in the westerlies is apparent in the observed record for January 1981, but none of the reanalyses replicate this spike. This might be an example of data quality control (e.g. gross error check based on Observation-minus-Forecast departures statistics) removing observational data during the assimilation processes. In the evolution from late 1997 to early 1998, the direct observations at most stations indicate double westerly peaks, but the second peak, in early 1998, was not observed at Abidjan (Fig. 6n). ERA-I and ERA-40 data follow this pattern and indicate a single peak but JRA-25, JRA-55, and MERRA display double peaks. Consequently the SD has a local maximum during this period. These results suggest that the error check procedures or assimilation schemes employed in the different centers may respond differently to anomalous single station observations.

The correspondence between periods of availability of radiosonde data and SD values among the reanalyses is sometimes unclear, such as for at Seychelles (Fig. 6b), 1980–82 at Belem (Fig. 6l) and 1984–98 at Abidjan (Fig. 6n). However in general, the SD in Fig. 6 is smaller during the months when radiosonde observations are available, and the SD tends to be larger when they are not available. These results suggest that the observations at these individual stations are having a significant influence constraining the reanalyses of the zonal wind in the tropical lower stratosphere.

To investigate what affects the longitudinal SD variation from the upper troposphere to the lower stratosphere, Fig 7 shows the longitude-height cross-section of zonal wind averaged over the five reanalyses and their SD (both averaged over the 1979-2001 period), as well as the temporal correlation for the period 1979-2001 between the absolute value of the zonal wind (i.e., the strength of the zonal wind, $|[u]|$) and the SD. All panels show the 10°N–10°S average. In the upper troposphere Fig. 7a displays the familiar Walker circulation signal in prevailing zonal winds. Fig. 7a suggests the eddy components of the prevailing zonal wind at 70 hPa (and to some extent even at 50 hPa) can be regarded as part of the Walker circulation. Hamilton et al. (2004) showed in their general circulation model experiments, that below about 60 hPa, the deviation of the zonal wind from the zonal mean appears to be dominated by an extension of the tropospheric Walker circulation.

A zonally elongated region of large SD exists around 100–150 hPa, the upper part of the Walker circulation (Fig. 7b). This upper tropospheric SD is largest in the central Pacific, where few observational stations are located (Fig. 5a). It is interesting that the region of large SD around the central Pacific in the upper troposphere connects with that seen in the lower stratosphere.

The correlation between the absolute value of zonal wind and the SD among reanalyses shows relatively high positive correlation in the upper part of the Walker circulation in the eastern hemisphere, while the correlation is relatively low in the central Pacific (Fig. 7c). In the mean state, the eddy component in the tropical lower stratosphere might be associated with an extension of the Walker circulation, but 70hPa standard deviation is relatively small, and it seems not correlated with the strength of the upper tropospheric Walker circulation. The large SD from the upper troposphere to the stratosphere in the
central Pacific could be simply related to the fewer *in situ* observations available there. In the middle stratosphere, the correlation is negative, indicating that the reanalyses disagree the most when the magnitude of the zonal wind is weakest. This corresponds to large SD during the phase transition of the QBO, as shown in the next section.

### 3.3. Differences in the middle stratosphere

In this subsection we discuss the SD at 10–30 hPa where generally fewer radiosonde observations are available. At 20–30
15   hPa, the SD becomes more zonally uniform compared with results at 50–70 hPa (Fig. 4). However, relatively higher-density observations around the Indonesian maritime continent, as well as over the North and South American and African continents do act to reduce the SD locally (Fig. 5c and 5d). At 10 hPa, Singapore is the only station quite near the equator with high-density coverage, and several stations around 7–10°N in the western Pacific have data coverage of 40–80% (Fig. 5b).

Because the SD among reanalyses involves more zonally uniform structures in the middle stratosphere (Fig. 4), the zonally averaged SD is investigated first. Figure 8 shows zonal mean equatorial zonal wind in each reanalysis plotted together with the zonally averaged SD among reanalysis at 10, 20, and 30 hPa. Large differences are seen when the QBO phase changes from easterly to westerly. In addition, the SD is generally larger during the westerly phase than during the easterly phase of the QBO. The dependence of SD on the QBO phase is most pronounced at 10 hPa.

To further investigate this dependence, a zonal wind composite based on the phase of the QBO was computed for the FUB observations and for each reanalysis over 1979–2001. For the easterly-to-westerly composite, the month '0' is defined as the final month with a mean easterly value in the 10 hPa FUB data and month '+1' is the first month with a mean westerly value. Composite values are then computed for ±6 months around these zero months (month 0 in August 1979, January 1982, March 1984…) for the easterly-to-westerly phases; see Fig. 2a. A similar westerly-to-easterly composite was also computed
(with month 0 taken at June 1980, January 1983, June 1985…). We chose 10 hPa as presenting the biggest challenge for the reanalyses as the numbers of radiosonde observations are fewest and the SD among reanalyses is largest.

The QBO composites for the zonal wind at 10 hPa are shown in Figs. 9a–d. Results for the zonal wind at Singapore are shown in the left panels (a,c) while results for the zonal mean equatorial zonal wind are shown in the right panels (b,d). Through the easterly-to-westerly transition at Singapore (Fig. 9a), the composite zonal winds in most of the reanalyses have an easterly bias relative to the FUB observations particularly during months −1 to +3. This results in a delay in the time of the zero-crossing in the reanalyses relative to the FUB observations. The reanalyses differ considerably in terms of this bias, however. The delay is ~1.5–2 month in JRA-25, JRA-55 and ~0.5 month in ERA-40 and ERA-I. The MERRA zonal winds show good agreement with FUB data during the easterly-to-westerly transition. The zonal wind values in the reanalyses show better agreement with the FUB observations and closer agreement among themselves during the westerly-to-easterly phase transition (Fig. 9c).

In composites of the zonal mean zonal wind (Figs. 9b, d), during the six months following the transition date the maximum westerly in each reanalysis is slightly smaller than in the zonal wind over Singapore (~3–4 m s$^{-1}$ for ERA-40 and ERA-I, ~2–3 m s$^{-1}$ for JRA-25 and JRA-55, nearly the same for MERRA), while the maximum easterly is nearly same. In both phase transitions, the spread of reanalysis zonal mean zonal winds is larger than that over Singapore. It should be noted that the timing of both phase transitions in the zonal mean are generally the same as those seen just using results at Singapore (although there is a slight delay of ~0.5 months in the easterly-to-westerly transition of the zonal mean in the JRA-25 and JRA-55) .

Figures 9e and 9f show a QBO composite of the zonal mean equatorial SD among five reanalyses due to zonal mean and eddy components (square roots of the first and second terms on the right in Eq.(2), respectively) at 10 hPa. It is clear that the SD due to zonal mean components is large during the phase transition from easterly to westerly, while the SD does not change much during the 12 composite months during the westerly-to-easterly (Fig. 9e). The SD due to eddy components has larger values during the westerly than the easterly phase (Fig. 9f).

It is well known that the transition of the QBO from easterly-to-westerly is considerably faster than that from the westerly-to-easterly (e.g. Baldwin et al., 2001). In other words, the zonal wind tendency $|\partial u/\partial t|$ is larger during the easterly-to-westerly transition. As the QBO could not be simulated without data assimilation from each reanalysis model (except MERRA-2), one possible reason for the large SD in the easterly-to-westerly is weak forcing by resolved waves in the reanalysis model, leading to slow change of the zonal wind and resulting delay of the phase transition. The results of Fig. 9 suggest that the total observational constraints are insufficient to completely compensate for model bias, resulting in slow change of the zonal wind and delay of the phase transition. The another possibility is that during periods of weak zonal wind there will be delays in the zonal advective propagation of information introduced into the analysis system from observations at individual stations (notably Singapore). We are not able to definitely assess these two possible mechanisms, but the fact that the lag in the reanalysis winds is more pronounced in the more rapid easterly-to-westerly wind transition may favor the importance of the model bias over the slow advective propagation mechanism.

Each reanalysis model may not have sufficiently fine vertical resolution to represent completely the interaction between vertically propagating waves and the mean flow, which is thought to be crucial to the QBO dynamics (e.g. Baldwin et al.,

2001), because the vertical wavelengths of waves become smaller as they approach critical levels. Podglajen et al. (2015) show vertical profiles of meridional wind by radiosonde observation at Singapore and in ECMWF operational analysis and indicate that ECMWF does not adequately represent wave disturbances due to insufficient vertical resolution (Fig. 9 of their study). MERRA has the finest vertical resolution among five reanalyses in the 20–30 km layer, which may allow a better

representation of wave forcing in the strong vertical shears of the QBO. Kim and Chun (2015) investigate the momentum forcing of the QBO by resolved waves in reanalysis models, and show that MERRA has the larger net-resolved wave forcing than ERA-I, MERRA, and JRA-55 (Fig. 2 in their study). They also show that mean residual vertical velocity in the tropics, which generally acts in the opposite sense to wave forcing in driving the zonal wind acceleration (e.g. Kawatani et al., 2011), is much smaller in MERRA than in other reanalyses (Fig. 5a in their study). These are possible reasons why the MERRA

shows faster transitions both in easterly-to-westerly and westerly-to-easterly transitions, compared with those in the other reanalyses.

Next we will discuss the representation of eddy components of the wind in the reanalysis datasets. Figures 10a–e show $u'$, the deviation of the monthly mean zonal wind values from the zonal mean, at 10 hPa for each reanalysis in January 1999, during the westerly phase of the QBO. In all the reanalyses the eddy component near the equator appears to be dominated by

zonal wavenumber 1 and 2 quasi-stationary planetary waves propagating from mid-latitudes, but the patterns differ somewhat among the reanalyses. For example, ERA-40 shows a large, local, positive anomaly to the east of the Indonesian maritime continent, while JRA-55 does not. Each reanalysis shows a different shape for this positive anomaly. In addition, the negative anomalies around the Bay of Guinea and central Pacific are also different.

Figure 10f shows one example of the eddy components of the zonal wind in January 1996, during the easterly phase of the

QBO in ERA-I. The eddy components are very small over the equator, which is expected as stationary planetary waves cannot propagate into the equatorial mean easterlies. Figure 10g depicts the longitudinal variations of each reanalysis zonal wind at 10 hPa over the equator in January 1996 and 1999. It is clear that the spread of the zonal wind among reanalyses is larger in the westerly phase, and longitudinal variation of the zonal wind is smaller in the easterly phase. We have checked other periods and found the monthly mean eddy structures in the middle stratosphere associated with quasi-stationary

planetary waves are qualitatively similar among years (not shown). The large contribution to SD from eddy components during the westerly phase of the QBO (Fig. 9f) mainly results from different representation of mid-latitude planetary waves propagating into the equator among reanalyses.

### 3.4. Difference depending on the QBO phase

In this subsection, we investigate the dependence of the SD among the reanalyses on the QBO phase. We have computed the

SD separately for the westerly and easterly phases of the QBO defined by the sign of the five reanalyses averaged zonal mean zonal wind at the equator (i.e. defined as $[\bar{u}]_{eq} > 0$ for the westerly and $[\bar{u}]_{eq} < 0$ for the easterly phase; subscript $eq$ indicates the equator). Figures 11a,b are maps of the SD among reanalyses at 10 hPa during the westerly and easterly phases

of the QBO for 1979–2001. Figs. 11c-g plots the zonal variation of the SD averaged over 10°S–10°N at each height in the easterly and westerly QBO phases. At 10hPa, large SD values in the westerly phase are seen in the Pacific, and from the Atlantic to Africa, where representations of quasi-stationary planetary waves differ significantly among the reanalyses (see Fig. 10). On the other hand, in the easterly phase, the SD is more zonally uniform. It is interesting that both in the westerly

and easterly phases, the SD declines significantly around the longitudes near Singapore. At 10 hPa the Singapore station provides the only continuous record of *in situ* data near the equator (Fig. 5b), and the result in Fig. 11c suggest that observations from this one station act as a strong constraint on the reanalyses.

The overall larger SD in the westerly QBO phases, as compared to the easterly QBO phases, especially over the Pacific, is still clear at 20 hPa (Fig. 11d). However, this feature is not apparent at 30 hPa (Fig. 11e), suggesting that the effects of quasi-

stationary planetary waves become much smaller at 30 hPa. At 50 hPa, the SD in the westerly and easterly phases is nearly identical (Fig. 11f). At 70 hPa the longitudinal variation of the SD in the westerly phase is similar to that in the easterly phase (Fig. 11g). This indicates that the QBO phase is not very important for SD zonal variations in the lower stratosphere.

## 4. Evolution of the differences among reanalyses with time

In this section, we discuss how the SD among reanalyses changes with time as the *in situ* and remote data sources available

for the reanalyses evolve. We will show the results from the four reanalyses datasets (ERA-I, JRA-25, JRA-55, and MERRA) that exist over the entire 34 year period 1979 to 2012. Figures 12a–c are maps of the SD averaged over 50–70 hPa for three 11-year intervals: 1979–1989, 1990–2000, 2001–2011. The SD among reanalyses reduces significantly with time, but it is interesting that geographical distributions of the SD remain quite similar over these three periods.

The left panels in Fig. 13 show time variations of zonal mean SD among the four reanalyses at each height averaged over

10°S-10°N. At 70 hPa and 50 hPa (Figs. 13j, m), the overall level of SD reduces gradually in the 1980s and 1990s, and a clear drop is found around 1998 at 50 hPa. At 10–30 hPa (Figs. 13a,d,g), the SD also reduces with time, but a sudden drop is not apparent.

The middle panels in Fig. 13 plot the same quantity as the left panels, but for the SD due to zonal mean and eddy components separately. At 70 hPa (Fig. 13n), eddy components are comparable to zonal mean components for all periods,

and both components reduce gradually with time until ~2000. The typical level of SD in both components remains fairly constant after 2000. At 50 hPa, the drop of SD in the zonal mean components around 1998 is pronounced; the eddy components also appear to become somewhat smaller after ~1998.

At 10–30 hPa, zonal mean contribution to SD always dominates, but gradually reduces with time (Figs. 13b,e,h). However, throughout the record this quantity spikes generally near a particular phase of the QBO (i.e. during the easterly-to-westerly

phase transition). The eddy contribution to SD also reduces slightly with time, but not dramatically.

One possible reason for reduction of the zonal wind SD over time would be upgrading of satellite radiance observations. From 1979 to 2006, the TOVS {TIROS (Television InfRrared Operational Satellite) Operational Vertical Sounder} Stratospheric Sounding Unit (SSU), and Microwave Sounding Unit (MSU), were available. After May 1998, data from the

Advanced Microwave Sounding Unit (AMSU) became available. The advent of these new satellite measurements is a possible reason for the decline in the SD around 1998.

Figure 14a shows the time-height section of zonal mean SD of temperature among the four reanalyses. A large SD is found around 30–50 hPa, and this SD reduces significantly after ~1998. The JRA team has reported on stratospheric temperature issues with JRA-25 and discussed the cause of the biases (Onogi et al. 2007). Fujiwara et al. (2015) explained that the radiative scheme used in the JRA-25 forecast model has a known cold bias in the stratosphere, and the TOVS SSU/MSU measurements do not have a sufficient number of channels to correct the model's cold bias; after introducing the ATOVS AMSU-A measurements in 1998, such a cold bias disappeared in the JRA-25 data product. On the other hand, with JRA-55 using a new radiation scheme in the forecast model, the stratospheric temperature during the TOVS period has been much improved.

In light of the possible issues with the JRA-25 temperatures we show in Figure 14b the temperature SD excluding JRA-25 (i.e. including only ERA-I, JRA-55, and MERRA). When JRA-25 is excluded, the very large SD values over 20–50 hPa before 1998 seen in Fig. 14a are greatly diminished. Even so the temperature SD among the three reanalyses (Fig. 14b) does reduce with time, with an especially large reduction over the full record at 100 and 10 hPa. The right panels of Fig. 13 show the same quantity as the left panels, but for zonal wind SD among just the ERA-I, JRA-55, and MERRA reanalyses. The SD among the three reanalyses is generally smaller than that among four reanalyses and the dramatic change in zonal wind SD at 50 hPa in 1998 become much weaker with the removal of JRA-25 (Fig. 13l), but the time evolution looks very similar.

Figure 15 shows the evolution of the number of available monthly mean radiosonde observations over 10°S–10°N and 10–70 hPa. The number of radiosonde observations available generally increased with time at all levels. For example at 70 hPa (50 hPa), the mean number of station-months of wind observations was ~33.7 (30.1) in 1979–1989, ~37.6 (35.1) in 1990–2000, and 44.5 (43.0) in 2001–2011 (i.e. periods corresponding to those in Fig. 12). So, relative to 1979–1989, the number of radiosonde observations at 70 hPa (50 hPa) increased by 11.6% (16.6%) in 1990–2000 and 32.0% (42.9%) in 2001–2011.

On the other hand, the zonal wind SD among the three reanalyses at 10°S–10°N (right panels in Fig. 13) and 70 hPa (50 hPa) reduces 20.8% (20.4%) in 1990–2000 and 33.7% (36.3%) in 2001–2011, compared with those in 1979–1989. The abrupt drop around 1998 is not clearly reflected in the number of radiosonde observations at 50 hPa, implying that the AMSU satellite observations could have significant impact on reanalysis of the zonal wind in the tropical stratosphere (see Figs. 13j, l).

At 10–30 hPa, the number of radiosonde observations increased significantly from 1979 to 2012, which should contribute to a reduction of the zonal wind SD with time at these altitudes, in addition to having more satellite data available. The same 10 hPa QBO composites as shown in Figs. 9a-d but during 1998–2012 are illustrated in Fig. 16. Here the zonal wind in ERA-40 is excluded (ERA-40 data exist only up to August 2002) but MERRA-2 is included (the unrealistic features noted earlier in MERRA-2 are not present after 1998). The general features are same as those in 1979–2001 such as an apparent delay in easterly-to-westerly transitions in JRA-25 and JRA-55 compared with the FUB observation and closer agreement among reanalyses in westerly-to-easterly transitions. It should be noted here that in these reanalyses the 10hPa QBO in 1998–2012

are closer to the FUB observations both during easterly-to-westerly and westerly-to-easterly transitions than during 1979–2001 (Fig. 9). In addition, the spread of reanalysis zonal winds become much smaller in the later period, especially for the zonal mean wind. Figure 16 also shows that the MERRA-2 reanalyses display an easterly-to-westerly phase transition at Singapore that is even more rapid than in the direct balloon observations. These results may indicate that the gravity wave sources in MERRA-2 are now excessive (Coy et al. 2016).

Satellite radiance data will presumably affect the assimilated temperatures in the stratosphere, but will also have some influence on the wind (cf. Iida et al., 2014) mainly through the use of multivariate background error covariance matrix at analysis step, even though the coupling of temperature and wind may be somewhat weak as the Coriolis parameter is small near the equator. Thermal wind shears will be very sensitive to small errors in observed temperatures. After 2000 the amount of available satellite data increased greatly (e.g. Kobayashi et al., 2015), and the contributions of satellite radiance data to better representation of the tropical winds presumably also increased. However, it is hard to quantify the relative roles of global satellites and *in situ* radiosonde observations in reducing the zonal wind SD among reanalyses.

Finally, figures 17a–d are the horizontal maps at 70–50 hPa SD among the three most modern reanalyses (ERA-I, JRA-55, and MERRA) illustrated for 2001–2011, when the SD is smallest (Fig. 12). Here, we also calculated SD between ERA-I and JRA-55, ERA-I and MERRA, and JRA-55 and MERRA, separately. The SD becomes larger when JRA-55 data are included, and the SD between ERA-I and MERRA is smallest. However, in any case, the geographical patterns of the SD are always similar, independent of the reanalysis or period selected (see Figs. 4, 12, and 17). These results indicate the importance of zonal wind observations by local radiosondes for better representation of the tropical zonal wind in the lower stratosphere in reanalysis datasets. Just for reference, the same figure, but between MERRA and MERRA-2, is shown in Fig. 17e. The geographical pattern is also similar, even though the MERRA-2 assimilation was notable in using a dynamical model that simulated a spontaneous QBO.

## 5. Summary and concluding remarks

This paper reports on a project to compare the representation of the monthly-mean zonal wind in the equatorial stratosphere among major global atmospheric reanalyses data sets. We mainly confined our analysis to 20°N–20°S and 10–70 hPa and to data starting in 1979 (i.e. after the availability of operational satellite temperature soundings). We compared the zonal wind in the reanalyses with the Free University of Berlin (FUB) data set for equatorial monthly-mean winds, which, for this period, were based on balloon observations at Singapore. We also compared reanalysis values with balloon observations at many other low latitude stations included in the IGRA.

All nine reanalyses data sets we examined display monthly mean zonal wind values at Singapore that are reasonably close to the directly observed values at all levels between 70 and 10 hPa (Fig. 2). However, the NCEP-1 and NCEP-2 analyses have significantly larger differences with the Singapore observations than do the other reanalyses. Notably, NCEP-1 and NCEP-2 display considerably weaker (as much as 10 m s$^{-1}$) easterly QBO extremes than is apparent in the balloon observations, particularly during the 1980s. MERRA-2 also stands out as an outlier in displaying large differences from FUB at 10hPa in

the 1980's and in November–December 1993. The NCEP-CFSR uses the ERA-40 winds in the tropical stratosphere from July 1981 to December 1998, and the differences between NCEP-CFSR and ERA-40 during this periodare nearly zero. Further analysis and discussion here was largely restricted to the other five reanalyses (ERA-40, ERA-I, JRA-25, JRA-55 and MERRA).

We characterized the degree of disagreement among the reanalyses by the standard deviation (SD) of the monthly mean zonal wind values in the five data sets. This measure is a function of height, latitude, longitude, and time. At each height the SD displays a prominent equatorial maximum, indicating the particularly challenging nature of the reanalysis problem in the low-latitude stratosphere where *in situ* observations are relatively sparse and the Coriolis parameter is small. The SD in the tropical stratosphere also depends significantly on both longitude and height (Fig. 4). At 50–70 hPa, a large SD is seen for

the Indian Ocean, to the east of the maritime continent, the central Pacific, and the Atlantic, showing clear zonally non-uniform structures. At 10–30 hPa, on the other hand, the SD becomes more zonally uniform.

At 70–50 hPa, the distributions of the SD are closely related with those of *in situ* radiosonde observations (Figs. 5e, f). The region of largest SD extends over wide zonal range in the central Pacific, where few observations are available. In the vicinity of individual radiosonde stations the SD at 70 hPa generally reduces in periods when observations are available

relative to periods with no observations (Fig. 6).

At each level there is a tendency for agreement to be best near the longitude of Singapore (a result that holds in all QBO phases) suggesting that the high quality Singapore balloon observations act as a strong constraint on all the reanalyses. At 10 and 20 hPa, the zonal mean SD has a clear dependence on the phase of the QBO (Fig. 8). Specifically the SD generally jumps to 5–10 m s$^{-1}$ for a few months around the transition from easterly to westerly. At 30 hPa, the difference is smaller

(almost always less than 5 m s$^{-1}$) but it still shows the same QBO dependence. The main discrepancy with FUB observations at these heights is a tendency for the zonal wind accelerations to be delayed in the all reanalyses, except MERRA, around the easterly-to-westerly transition (Fig. 9).

At 10 hPa, all the reanalyses show that the eddy component (i.e. the deviation of the zonal wind values from the zonal mean) in the monthly mean zonal wind near the equator is dominated by zonal wavenumber 1 and 2 quasi-stationary planetary

waves propagating from mid-latitudes (Fig. 10). Consistent with this, the eddy component of the zonal wind near the equator is small in months with zonal mean easterlies, and generally much larger in months with zonal mean westerlies. While the different reanalyses agree in broad terms about the structure of the eddy components in the middle stratosphere, there generally remains considerable disagreement among the reanalyses in the details of the quasi-stationary waves near the equator. At 50–70 hPa the SD distributions are not strongly affected by the phase of the QBO. The eddy features at these

heights, rather than being planetary waves from mid-latitudes, might be related to stratospheric penetration of the Walker circulation (Fig. 7).

Our measures reflecting the reliability of the reanalyses show systematic improvement over the 1979–2012 period considered. Specifically, the agreement of the reanalysis results with the Singapore data improves significantly with time, with average deviations from the observation reduced by almost half from that in the early 1980s to the 2010s (Fig. 2f). At

70–50 hPa, the magnitude of the SD among the reanalyses declines significantly over the period of record (Fig. 13), although the geographical pattern of differences (as measured by the SD of the five reanalysis values) is nearly constant (Fig. 12).

For any individual reanalysis, the same atmospheric dynamical model and the same assimilation procedures are used throughout the whole period, thus any improvement seen in the quality with time must be related to changes in the observational data used. The number of available *in situ* radiosonde observations increase with time at all levels (Fig. 15). In addition, satellite data streams vary throughout this period. The end of 1978 saw the introduction of the Stratospheric Sound Unit (SSU) which was flown on NOAA operational satellites for the next two decades (NOAA8 though NOAA14). We expect an important upgrade may have occurred in May 1998 when the temperature sounders on NOAA operational satellites were upgraded from the SSU to the Advanced Microwave Sounding Unit (AMSU). The satellite streams continued to show changes with the introduction of more AMSU instruments flying on other satellites. As well new sources of radiance data also appeared, such as the Atmospheric Infrared Sounder (AIRS) flown on the NASA Aqua satellite in 2002.

Doing assimilation experiments removing certain data sets from the analysis and exploring the impact of particular data sets are one possible way to investigate the data included. The effects of satellite datasets might be investigated by comparison between JRA-55 and JRA-55c (which excludes all satellite data as input). Kobayashi et al. (2014) compare the JRA-55 and JRA-55C reanalyses and indicates that the tropical zonal wind difference between the JRA-55 and JRA-55C is large in the upper stratosphere compared to that in the lower stratosphere, and the amplitude in JRA-55C is smaller than that of the JRA-55. More detail comparisons along these lines are reserved for future work.

The magnitude of the SD in the horizontal winds throughout the tropics would be worth highlighting for trajectory studies given that there is some inclination to assume that the key the inter-reanalysis differences for chemical transport are in the vertical velocities. Abalos et al. (2015) have recently compared the stratospheric Brewer-Dobson circulation among MERRA, ERA-I and JRA55, while Miyazaki et al. (2015) did a somewhat similar study with 6 different reanalyses. Abalos et al. find quite large (~40%) differences among the reanalyses in the overall strength of the BD circulation and, that there is a large spread among the estimates of the fraction of variance in mean vertical motion explained by the QBO. Coy et al. (2006) found that MERRA-2 which internally generates the QBO has reduced the zonal wind analysis increments compared to MERRA, so that the QBO mean meridional circulation can be expected to be more physically forced and more physically consistent

The difference among reanalyses using twice daily data should be much larger than our SD based on monthly mean data (cf. Baker et al. 2014). Padglajen et al. (2014) compare reanalyses winds with independent *in situ* observations performed along long-duration balloon flights. They report that ERA-I and MERRA represent similar disturbances associated with equatorial waves, but reanalyses depart from the balloon observations. As the present study focus on monthly mean field, our analyses ignore variability with shorter time scales.

Our study confirms that even with relatively few balloon stations, the high accuracy and high resolution wind measurements by *in situ* radiosondes have provided important constraints to reanalyses of circulation in the tropical stratosphere. Our analysis also shows that changes in both satellite and balloon data availability over the years have significantly affected the

reanalysis of zonal wind data for the equatorial stratosphere. This conclusion has a negative aspect as it suggests that using reanalysis products (even restricting the data to after 1978) does not enable a researcher to avoid all significant artificial trends. On the positive side, it seems that the changes in available data have led to continually improving reanalysis representation of the tropical stratosphere.

## 5  Acknowledgement

We thank Prof. M. A. Geller, Dr. A. Hertzog and three anonymous reviewers for constructive comments on the original manuscript. We also express our gratitude to S-RIP members for providing useful information about the reanalyses data. This research was supported by Grant-in-Aid for Scientific Research B (26287117) and Joint international Research (15KK0178) from the Japan Society for the Promotion of Science, and by the Environment Research and Technology Development Fund (2-1503) of the Ministry of the Environment, Japan. This research was also supported by the Japan Agency for Marine-Earth Science and Technology (JAMSTEC) through its sponsorship of research at the International Pacific Research Center and by NOAA through grant No. NA11NMF4320128. The GFD-DENNOU Library and GrADS were used to draw the figures.

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

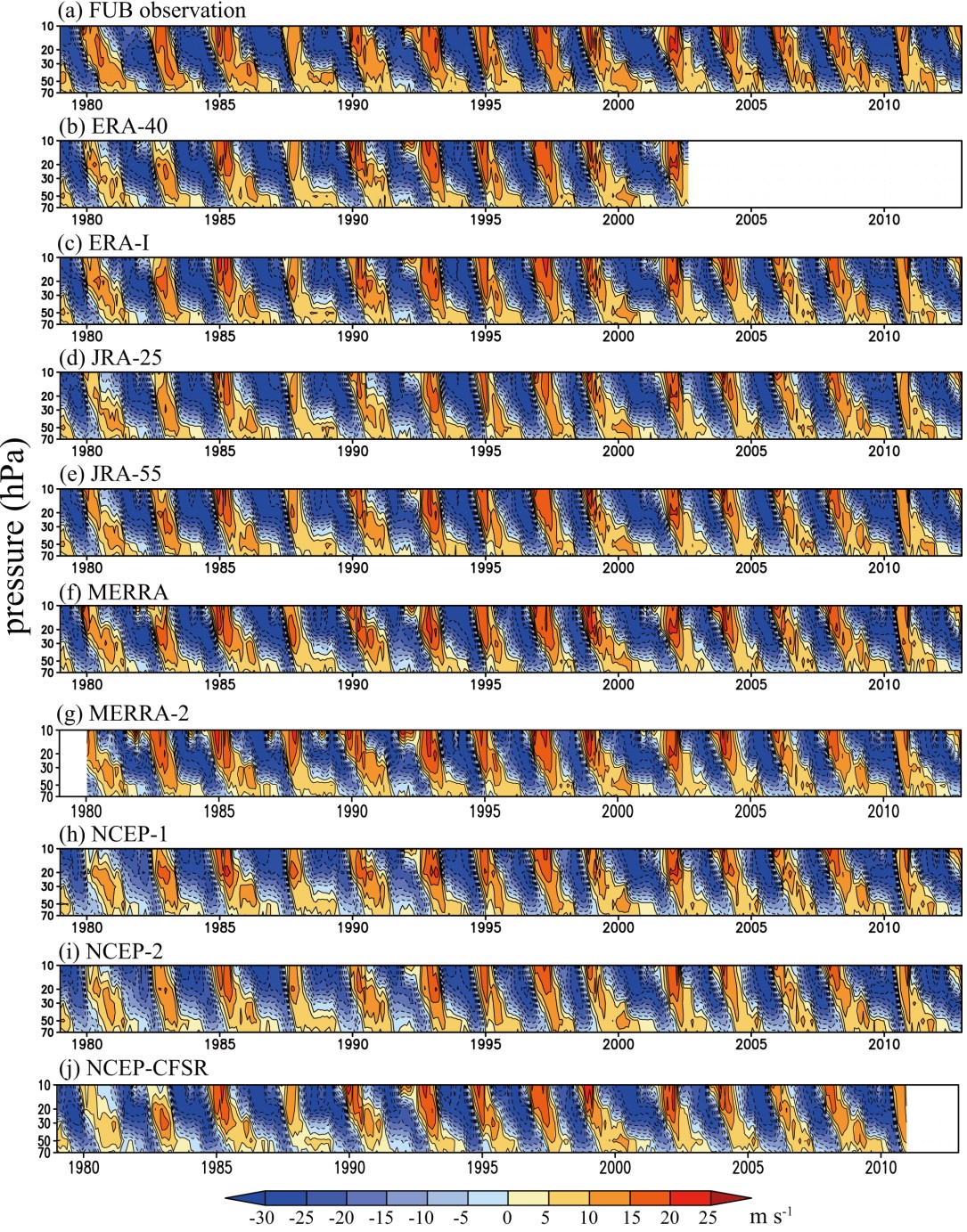

Figure 1. Time-height cross section of the FUB zonal wind and each reanalysis zonal wind over Singapore (104˚E, 1.4˚N) from 1979 to 2012. The color intervals are 5 m s$^{-1}$.

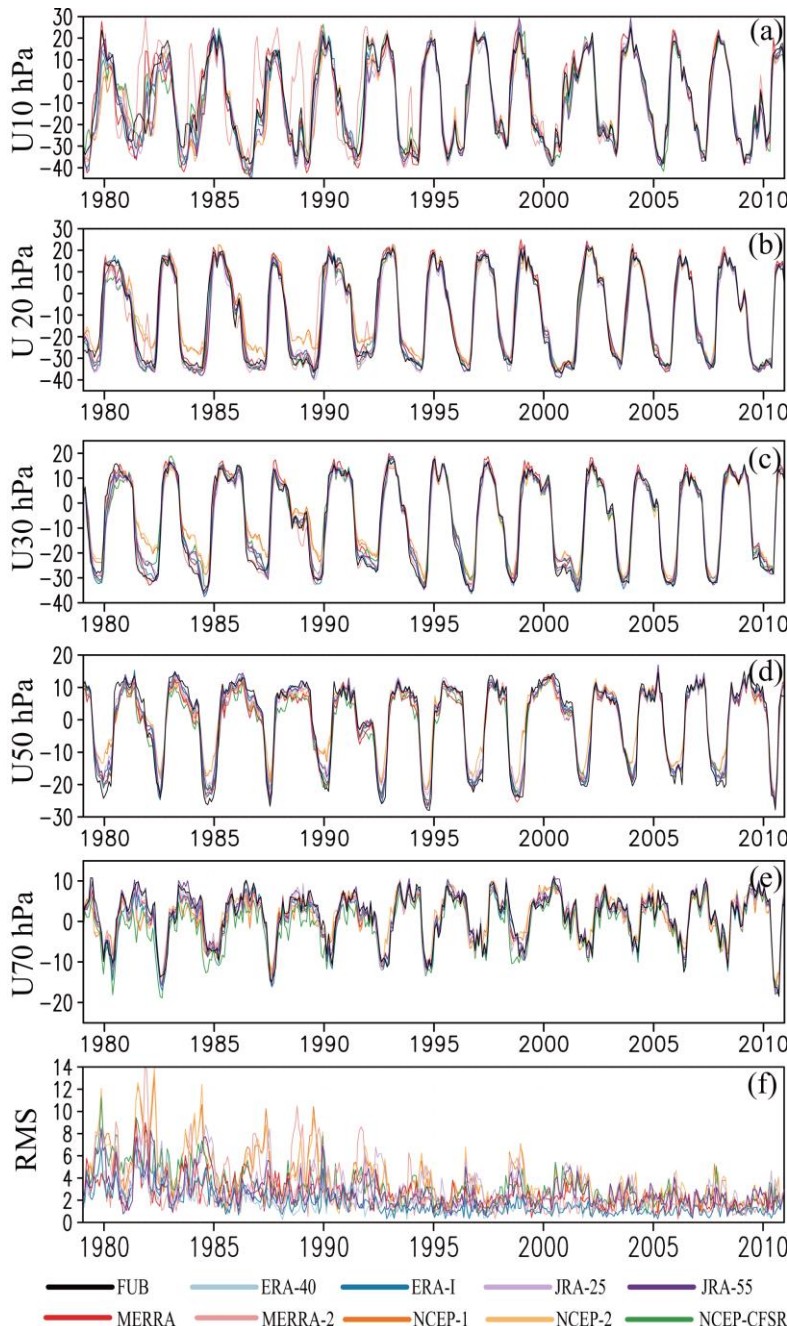

Figure 2. Time variations of the zonal wind over Singapore at (a) 10 hPa, (b) 20 hPa, (c) 30 hPa, (d) 50 hPa, and (e) 70 hPa for FUB observation (black), ERA-40 (light blue), ERA-I (blue), JRA-25 (light purple), JRA-55 (purple), MERRA (red),
5   MERRA-2 (pink), NCEP-1 (orange), NCEP-2 (yellow) and NCEP-CFSR (green); (f) Root mean square differences between the FUB and each reanalysis zonal wind, averaged at 70–10 hPa.

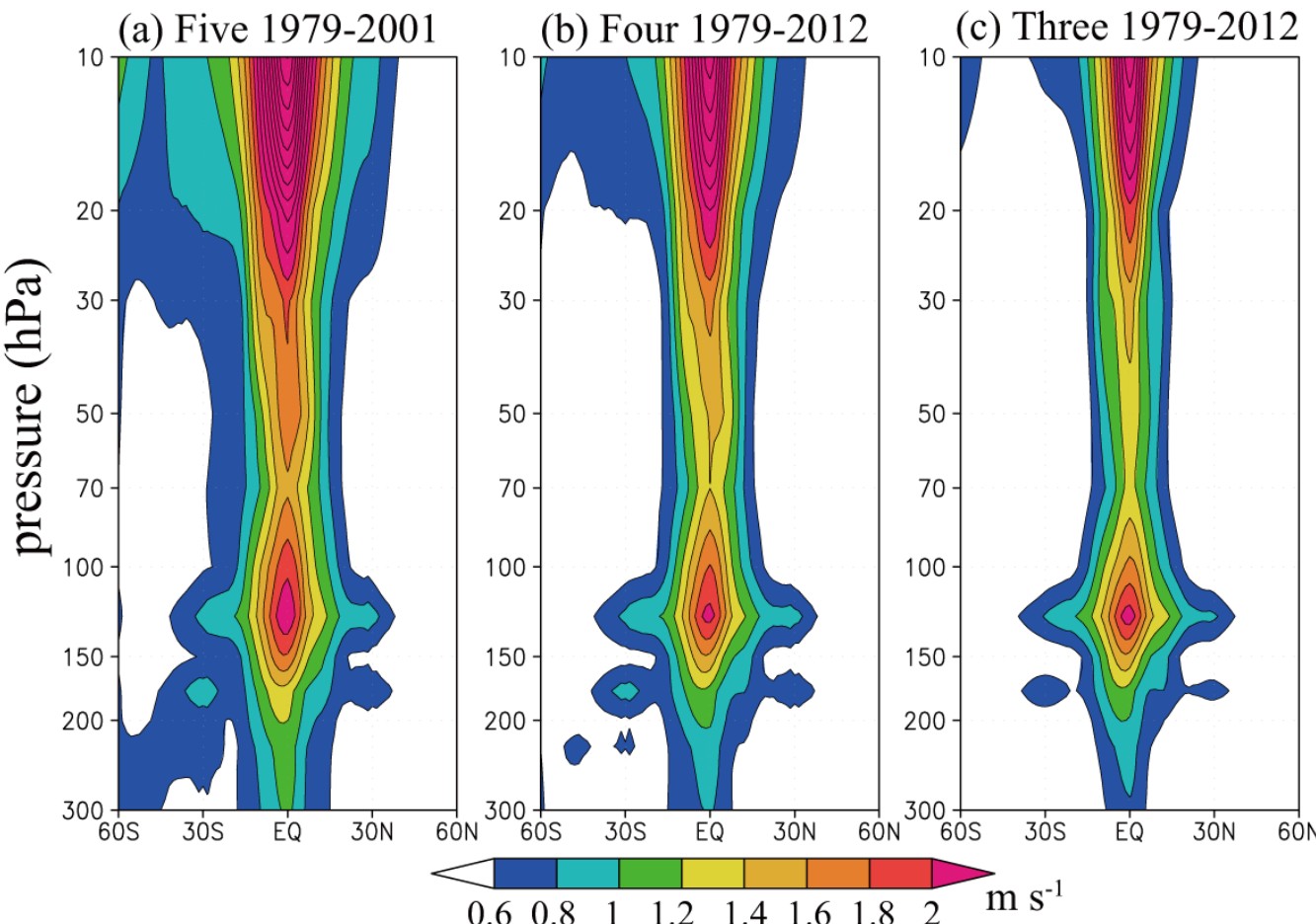

Figure 3. Latitude-height cross-sections of zonal mean and time mean of the standard deviation among (a) five reanalyses of ERA-40, ERA-I, JRA-25, JRA-55, and MERRA from 1979 to 2001, (b) four reanalysis of ERA-I, JRA-25, JRA-55, and MERRA from 1979 to 2012, and (c) three reanalysis of ERA-I, JRA-55, and MERRA from 1979 to 2012. The color intervals are 0.2 m s$^{-1}$ and shaded with values larger than 0.6 m s$^{-1}$.

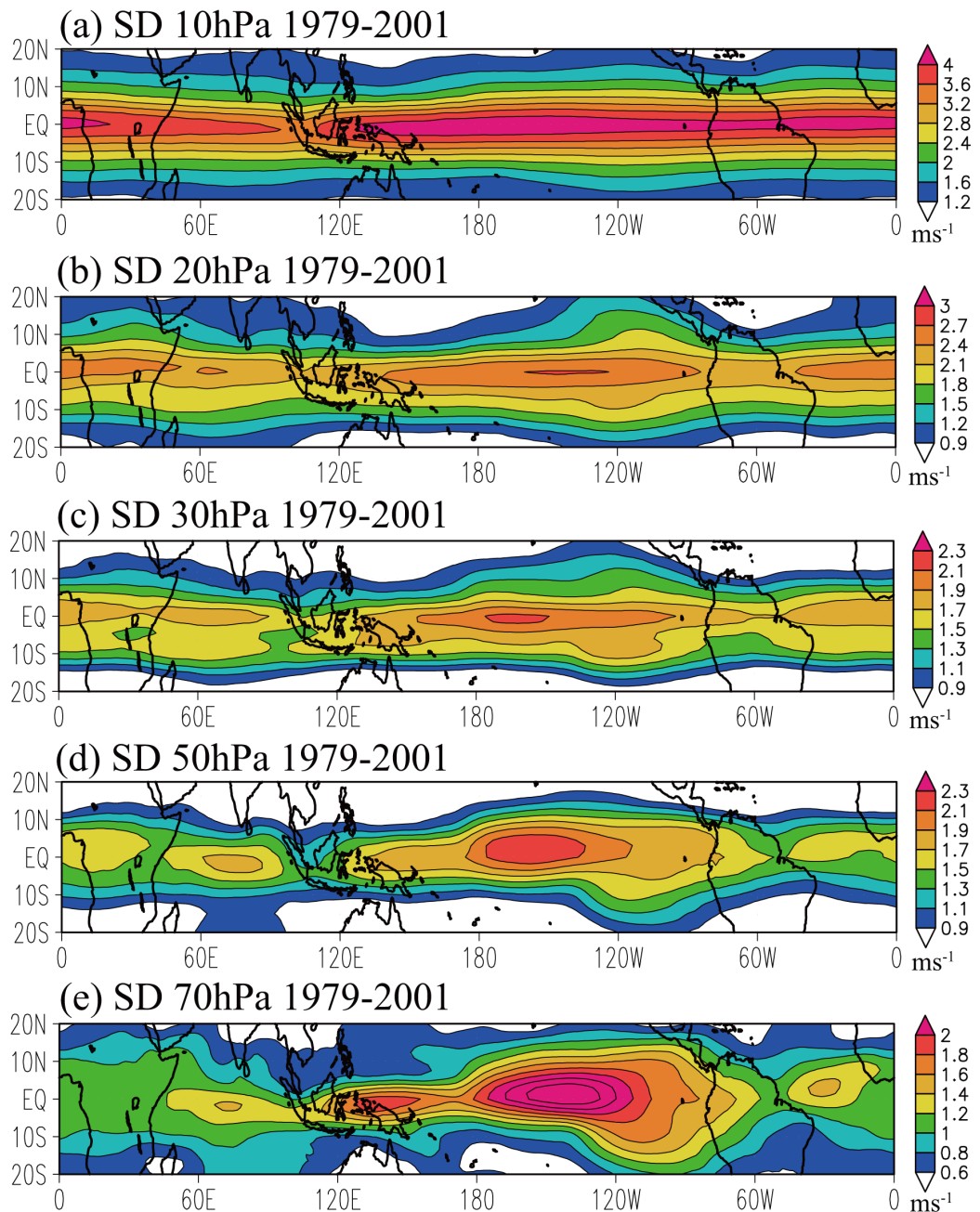

Figure 4. Horizontal maps of the standard deviation among the five reanalyses at (a) 10 hPa, (b) 20 hPa, (c) 30 hPa, (d) 50 hPa, and (e) 70 hPa: The color intervals are 0.4 m s$^{-1}$ for (a), 0.3 m s$^{-1}$ for (b), and 0.2 m s$^{-1}$ for (c-e). Color ranges are different among these heights.

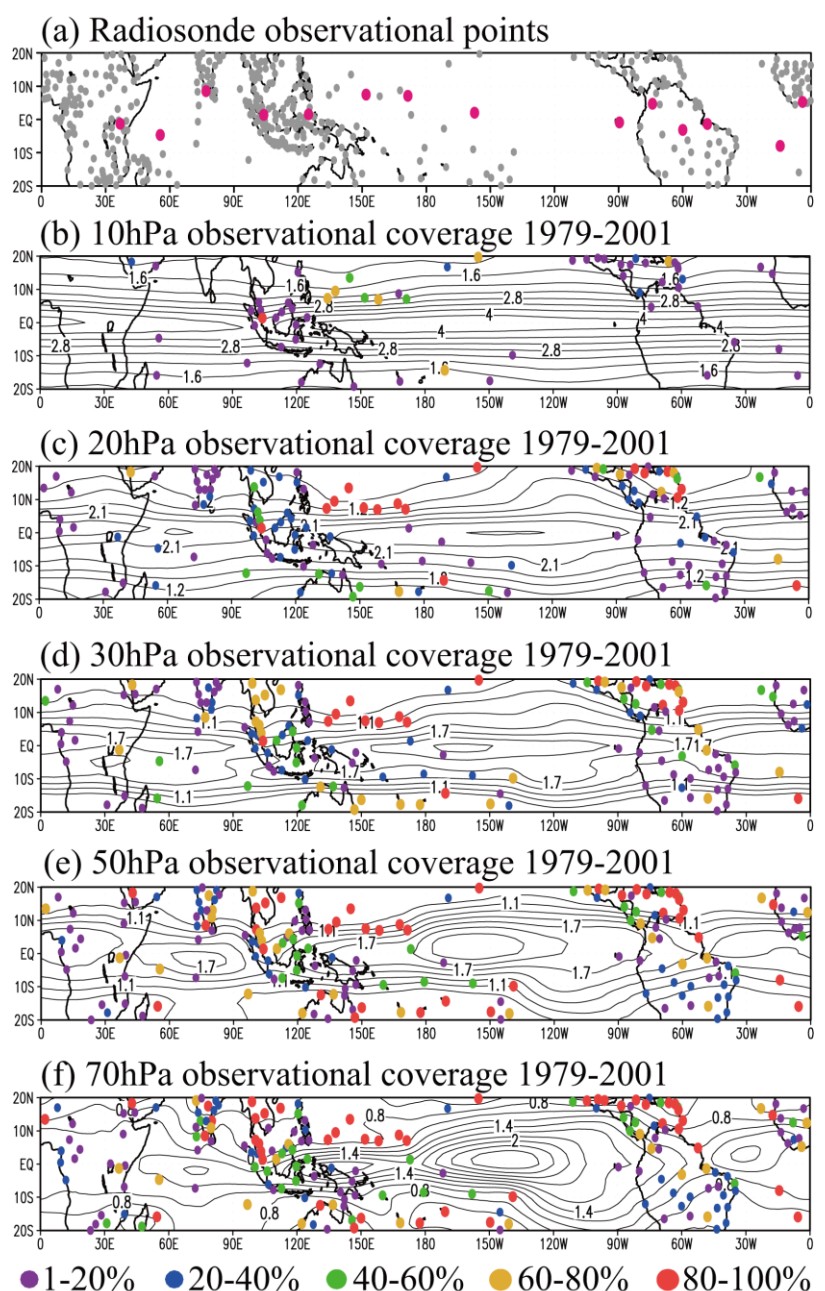

Figure 5. (a) Locations of all IGRA stations in the tropical region (magenta dots indicate the locations of the stations shown in Fig. 6). (b-f) IGRA stations with data coverage of (purple) 1−20%, (blue) 20−40%, (green) 40−60%, (yellow) 60−80% and (red) 80−100% at (b) 10 hPa, (c) 20 hPa, (d) 30 hPa, (e) 50 hPa, and (f) 70 hPa during 1979−2001. The contours show the standard deviation among reanalyses as shown in Fig. 4. The contour intervals at each height are the same as in Fig. 4.

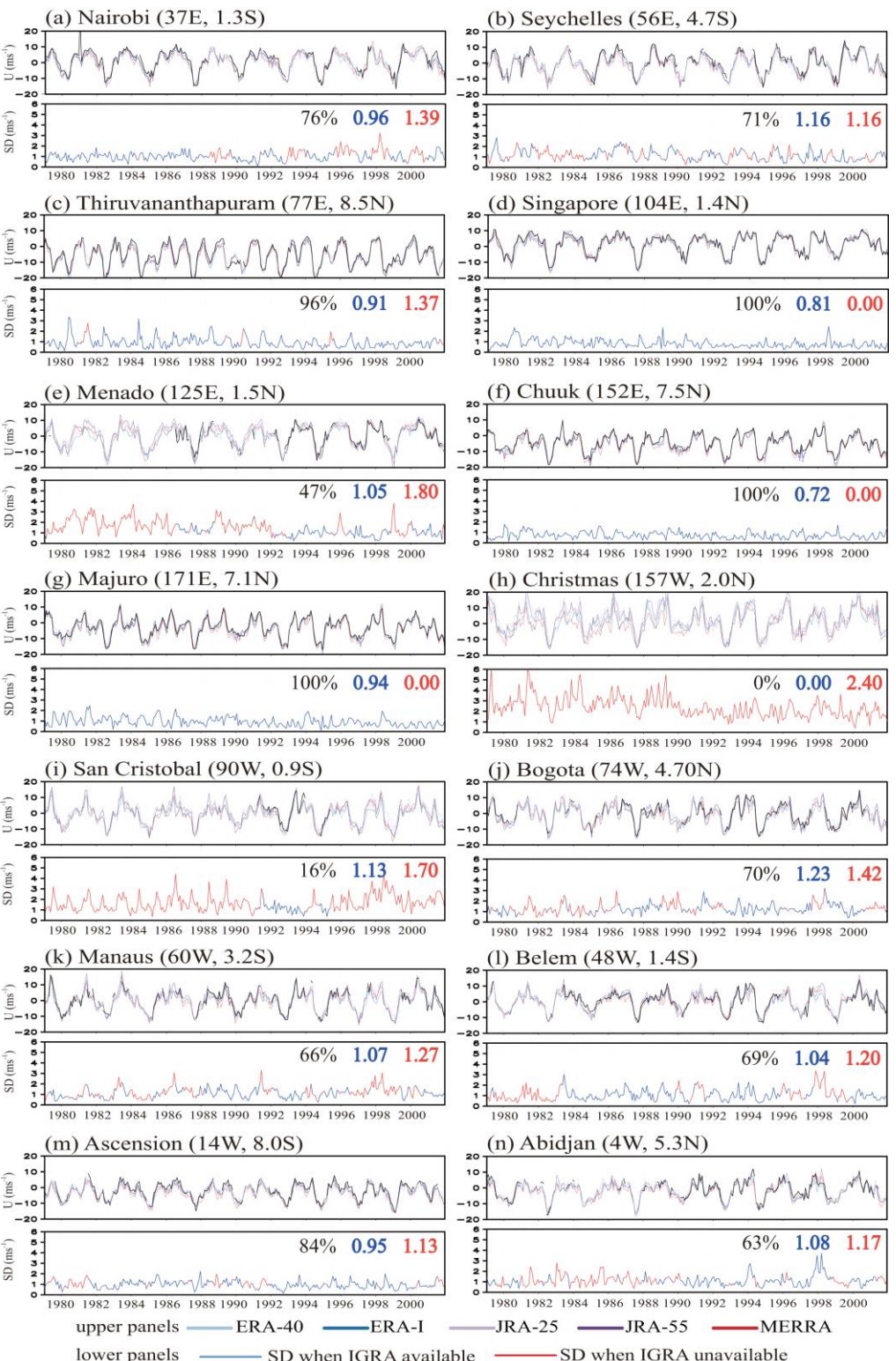

upper panels  —— ERA-40  —— ERA-I  —— JRA-25  —— JRA-55  —— MERRA

lower panels  —— SD when IGRA available  —— SD when IGRA unavailable

Figure 6. Upper panels show observed zonal wind (black) at each station and reanalysis zonal wind (color) at 70 hPa interpolated to each observational point from 1979 to 2001. Lower panels depict standard deviation among the five reanalyses at 70 hPa. Blue lines show the standard deviation during times when monthly-mean radiosonde data is available and red lines show the standard deviation during times when they are not. In the upper right corner of each bottom panel, the total data coverage (%) at each station and the average value of the standard deviation during periods with (blue number) and without (red number) radiosonde data are shown.

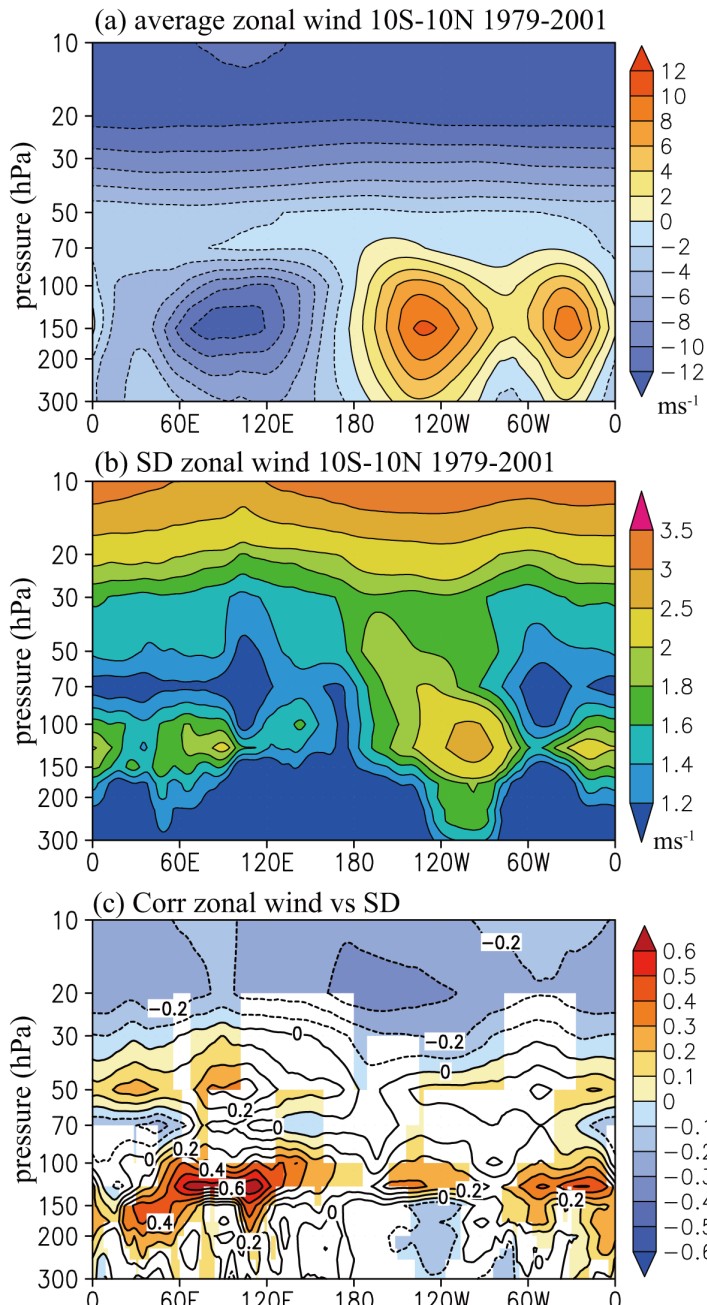

Figure 7. Longitude-height cross section of (a) averaged zonal wind in the five reanalyses, and (b) the standard deviation among reanalysis and (c) the correlation between absolute value of zonal wind and the standard deviation in 10˚N–10˚S averaged from 1979 to 2001. The color intervals are (a) 2 m s$^{-1}$, (b) 0.2 m s$^{-1}$ for values less than 2 m s$^{-1}$ and 0.5 m s$^{-1}$ for values more than 2 m s$^{-1}$, (c) 0.1, with statistical significance of ≥95% colored.

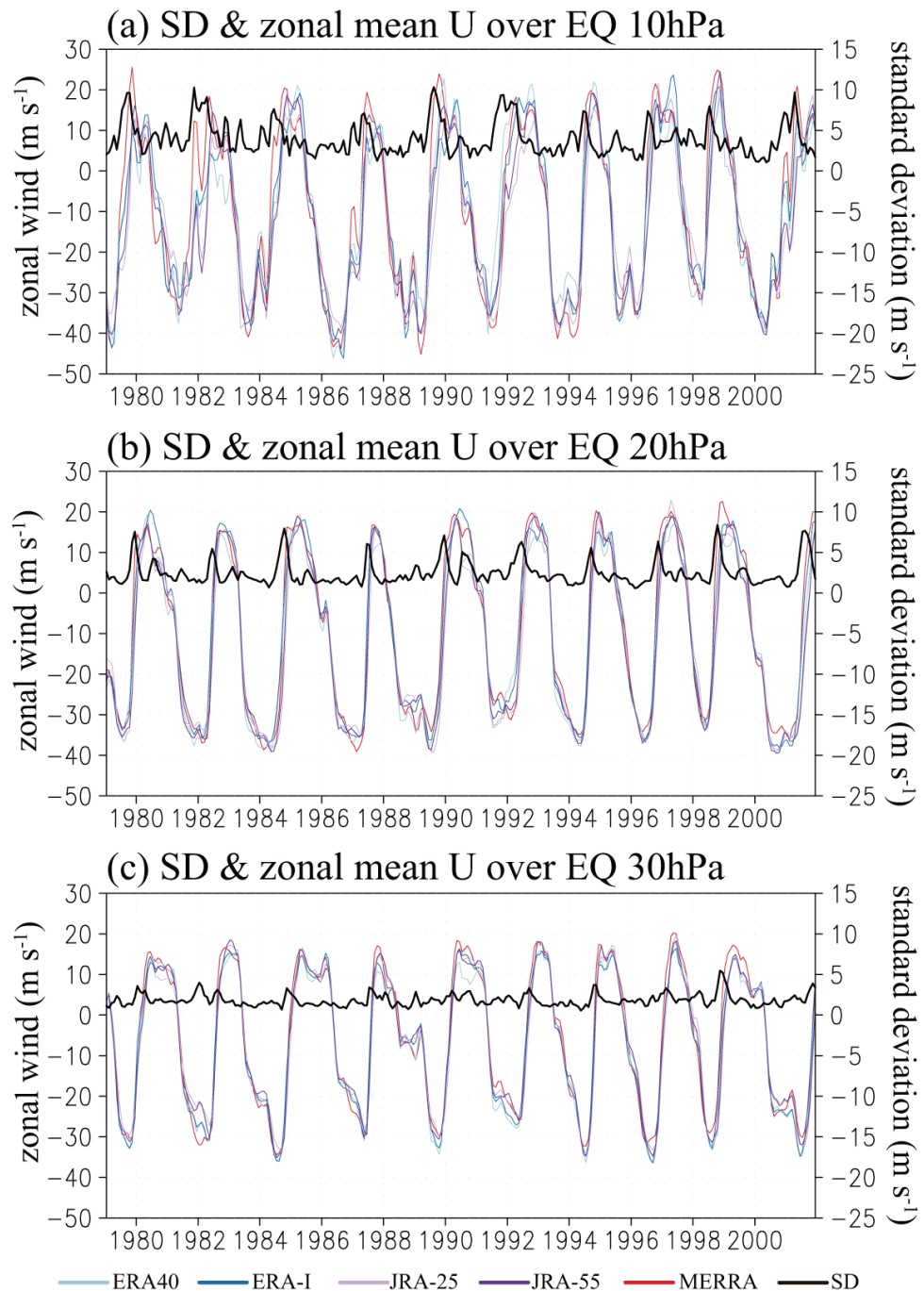

Figure 8. Zonal mean equatorial zonal wind in each reanalysis (colors) and zonally averaged equatorial standard deviation among reanalyses (black) at (a) 10 hPa, (b) 20 hPa, and (c) 30 hPa.

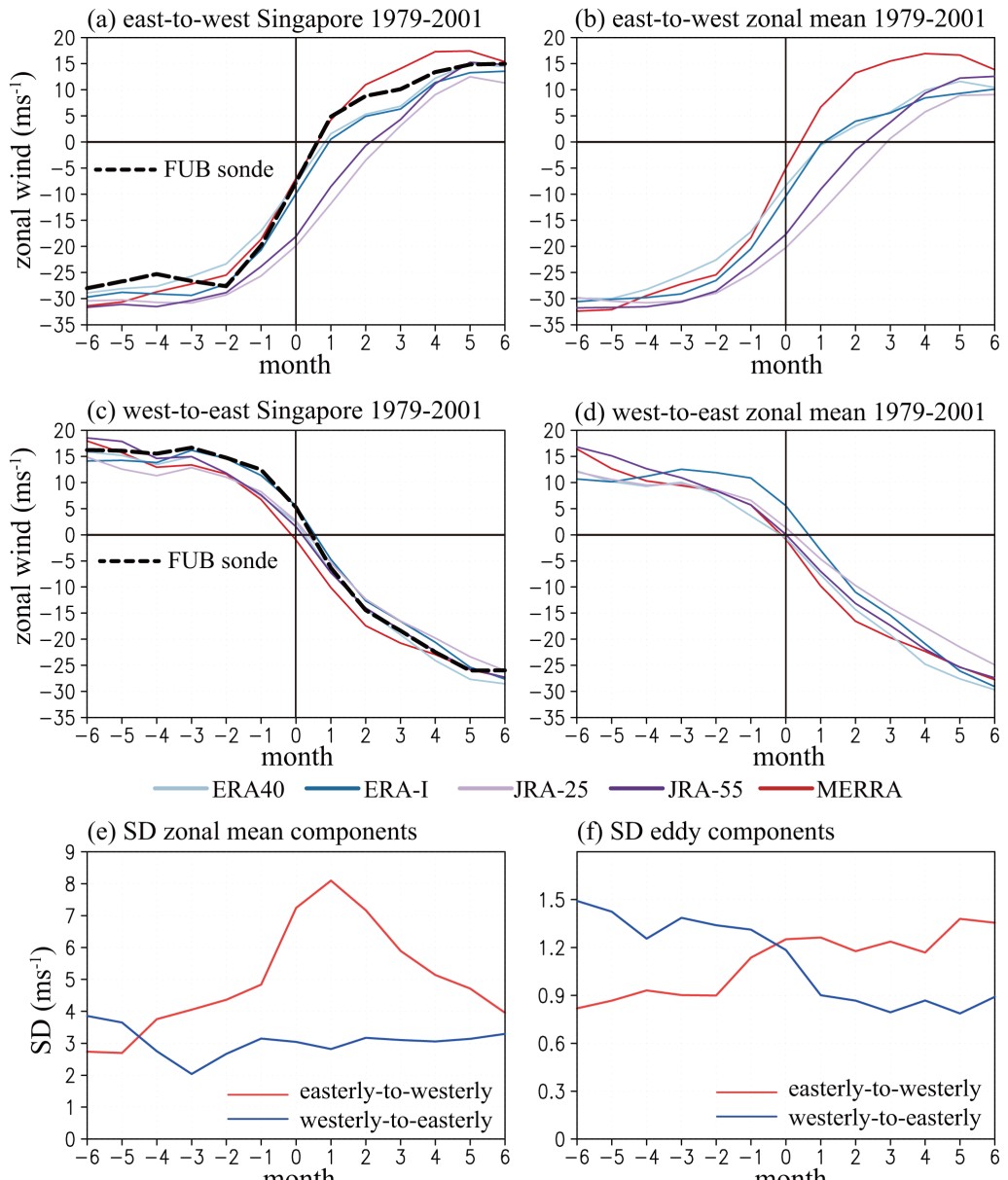

Figure 9. Top: Composite of the QBO in the zonal wind at 10 hPa during 1979 to 2001 where month 0 corresponds to (a, b) the easterly-to-westerly transition of the FUB zonal wind at 10 hPa and to (c, d) the westerly-to-easterly transition. Color lines show each reanalysis zonal wind, and black dashed lines depict FUB zonal wind at Singapore. Results for (a, c) the zonal wind at Singapore and for (b, d) the zonal mean equatorial zonal wind. Bottom: Composite of the zonal mean equatorial standard deviation due to (e) zonal mean and (f) eddy components during (red) easterly-to-westerly and (blue) westerly-to-easterly transitions at 10 hPa. Note the ordinate axes are different between (e) and (f).

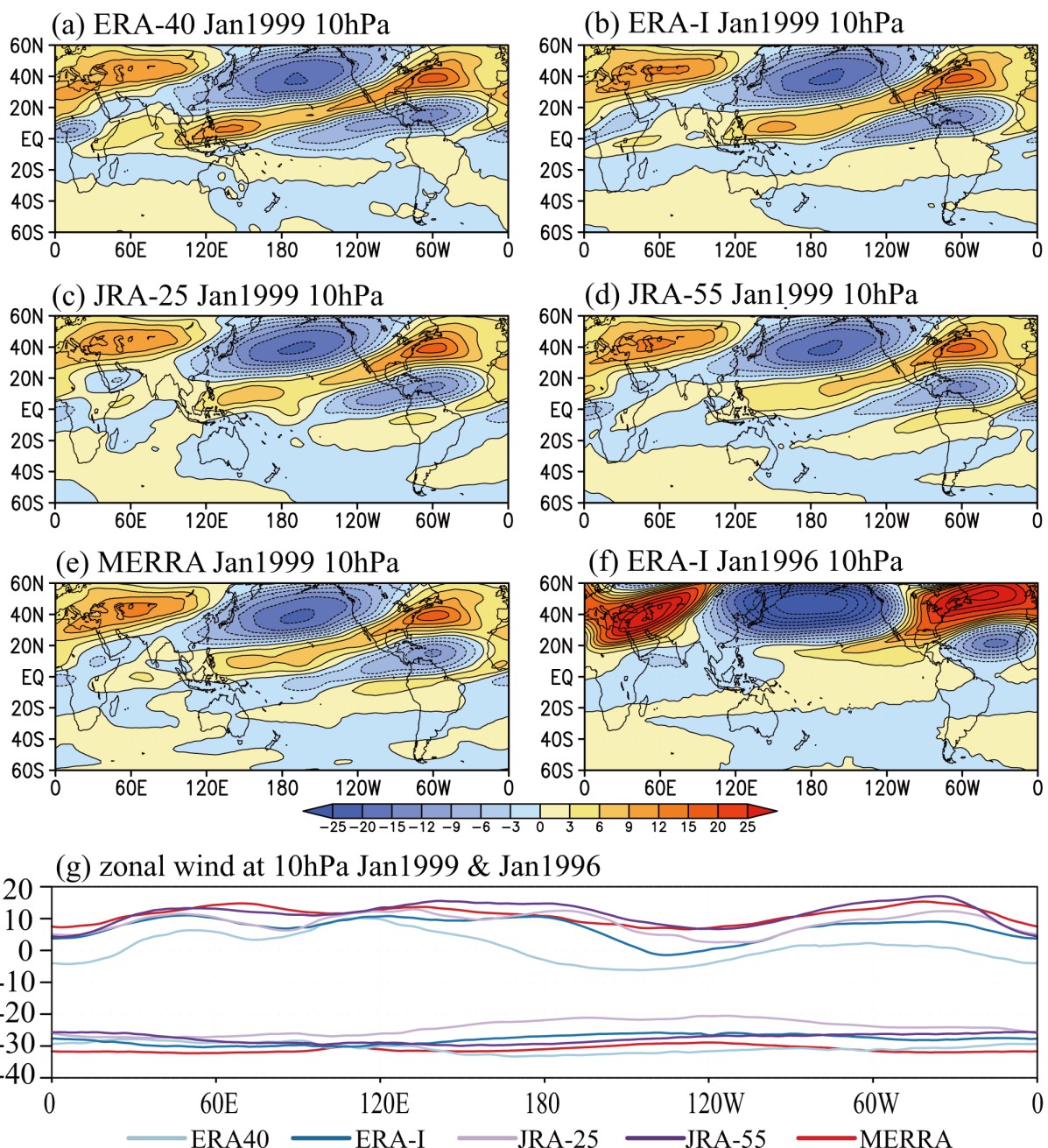

Figure 10. Deviation of the zonal wind values from the zonal mean ($u'$) at 10 hPa of (a-e) five reanalyses in January 1999 during westerly phase and (f) of the ERA-I in January 1996 during easterly phase of the QBO. The color interval is ±3, 6, 9, 12, 15, 20, or 25 m s$^{-1}$. (g) Longitudinal variations of the zonal wind of each reanalysis at 10 hPa over the equator in January 1999 and January 1996.

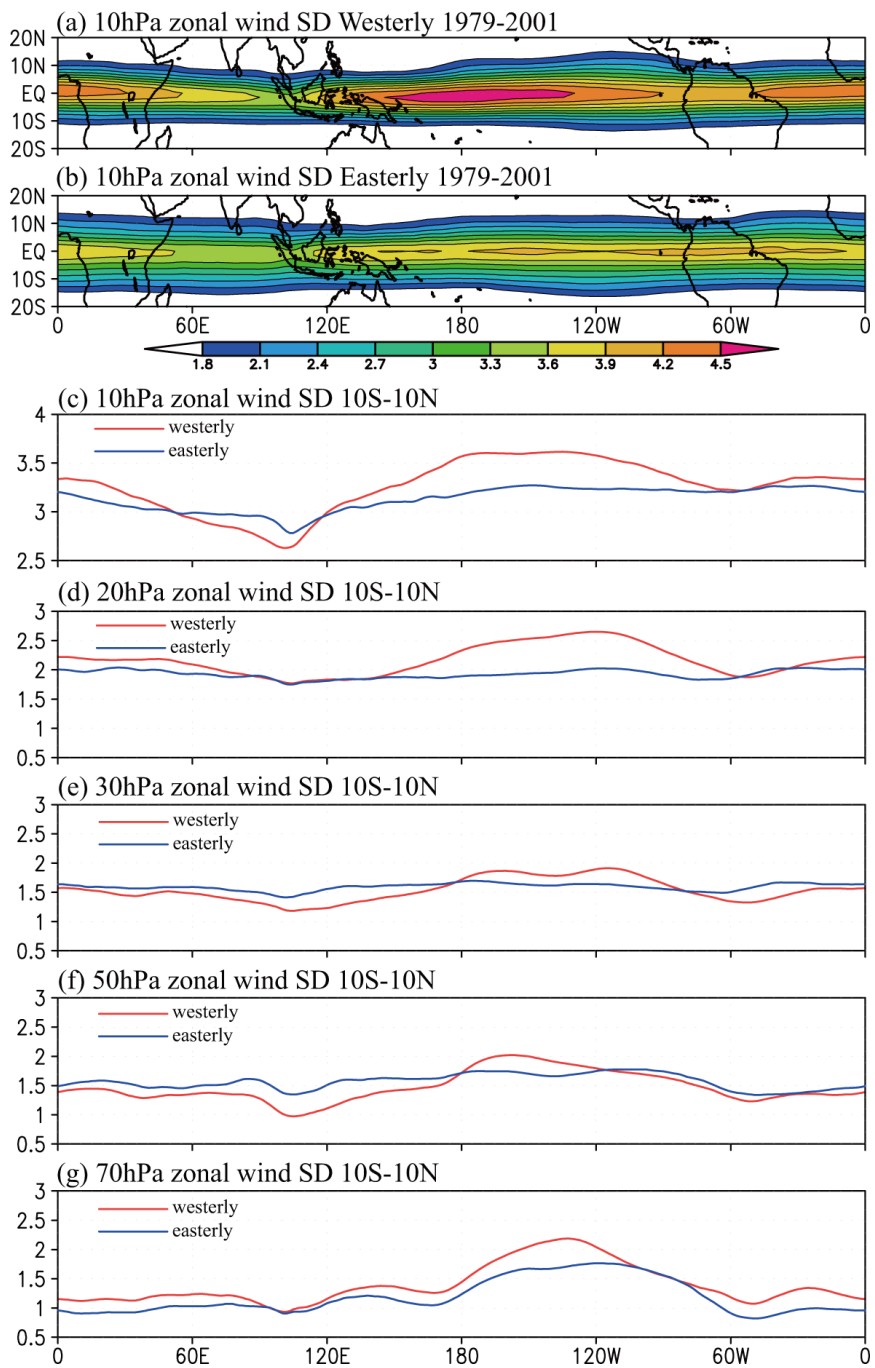

Figure 11. Horizontal distributions of the standard deviation among reanalyses at 10 hPa during (a) westerly and (b) easterly phase of the QBO averaged in the period 1979–2001. The color interval is 0.3 m s$^{-1}$. Longitudinal variations of the standard deviation at 10˚S–10˚N in (red) westerly and (blue) easterly phase at (c) 10, (d) 20, (e) 30, (f) 50, and (g) 70 hPa.

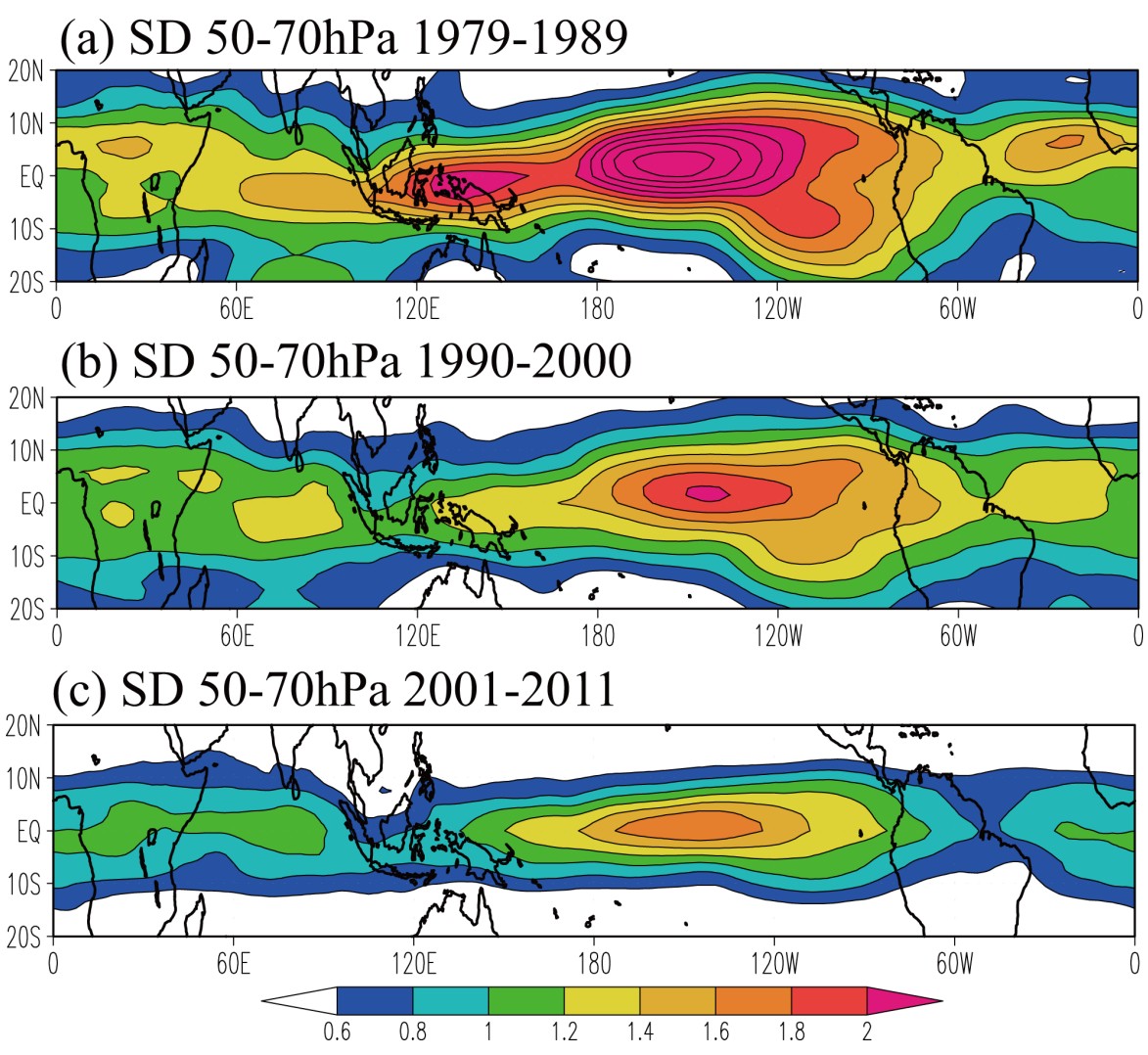

Figure 12. Standard deviation among four reanalyses (ERA-I, JRA-25, JRA-55, and MERRA) at 50-70 hPa for each 11-year mean: (a) 1979–1989, (b) 1990–2000, and (c) 2001–2011. The color intervals are 0.2 m s$^{-1}$.

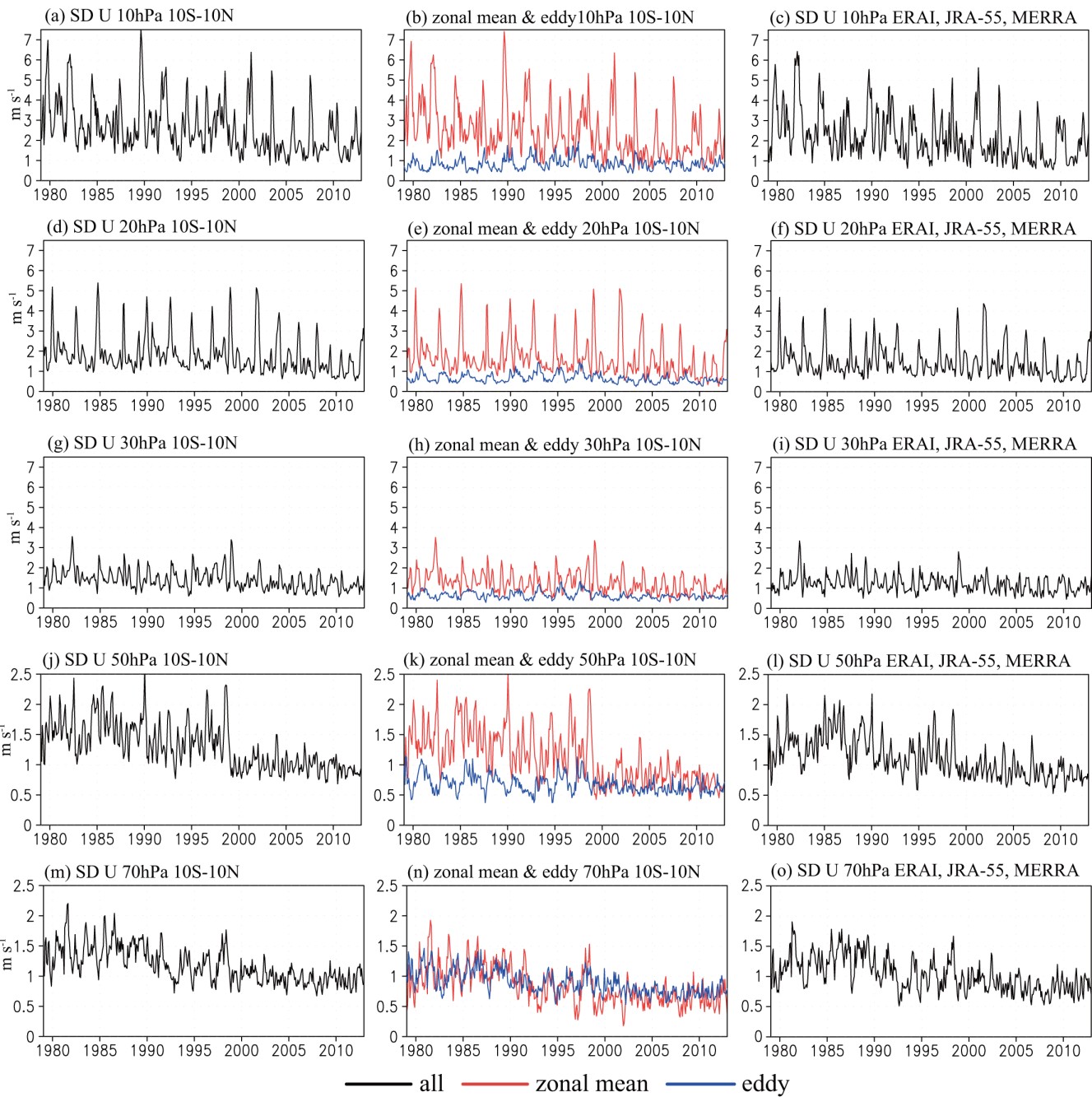

Figure 13. Time variations of zonal mean standard deviation of (black) all, (red) zonal mean, and (blue) eddy components among (left and middle) four reanalyses (ERA-I, JRA-25, JRA-55, and MERRA), and among (right) three reanalyses (ERA-I, JRA-55, and MERRA) from 1979 to 2012 at 10 hPa, 20 hPa, 30 hPa, 50 hPa, and 70 hPa in 10˚S-10˚N.

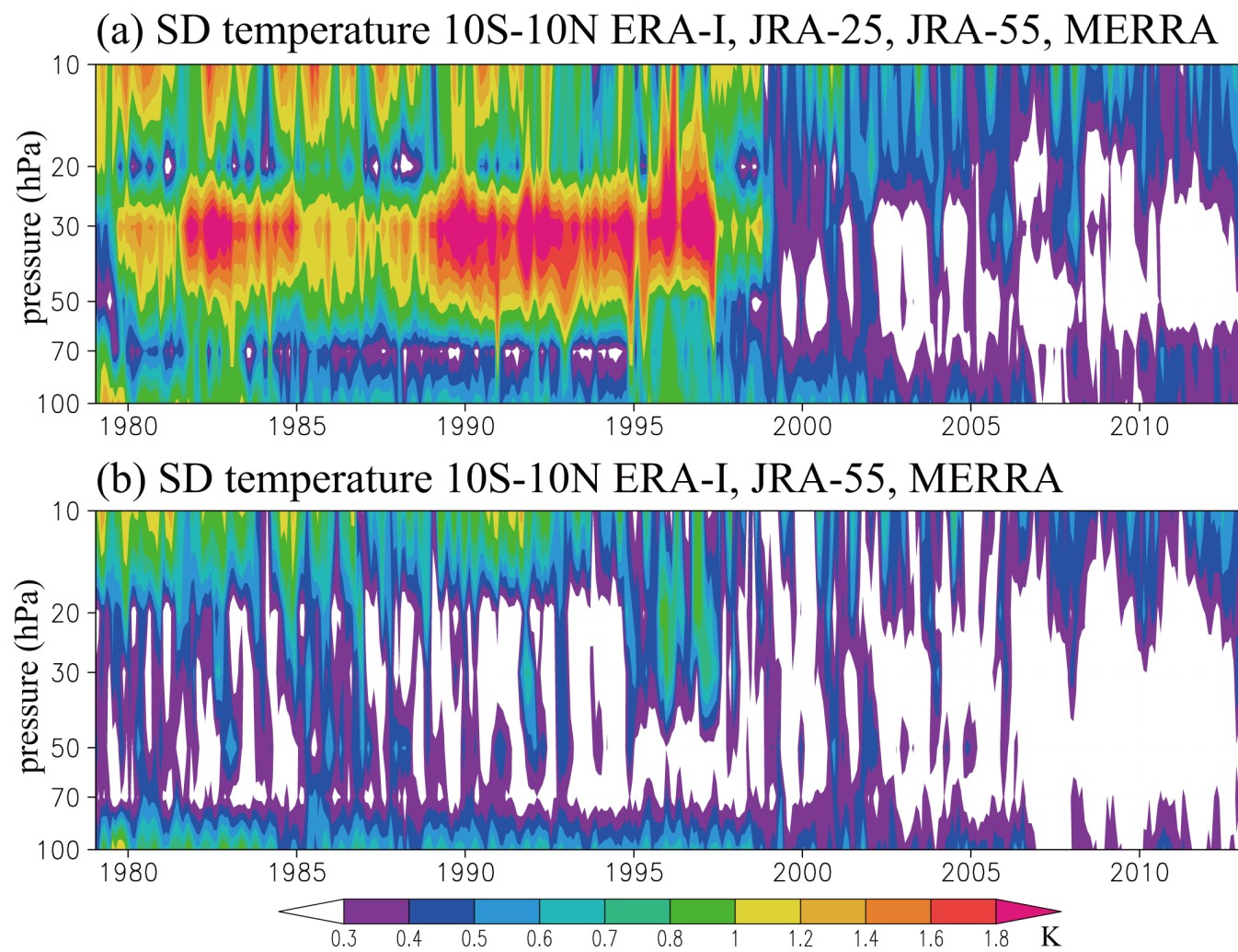

Figure 14. Time-height cross sections of zonal mean standard deviation of temperature among (a) four reanalyses (ERA-I, JRA-25, JRA-55, and MERRA), and (b) three reanalysis (ERA-I, JRA-55, and MERRA) at 10°S–10°N from 1979 to 2012. The color intervals are 0.1 K for values less than 0.8K and 0.2 K for values more than 0.8K.

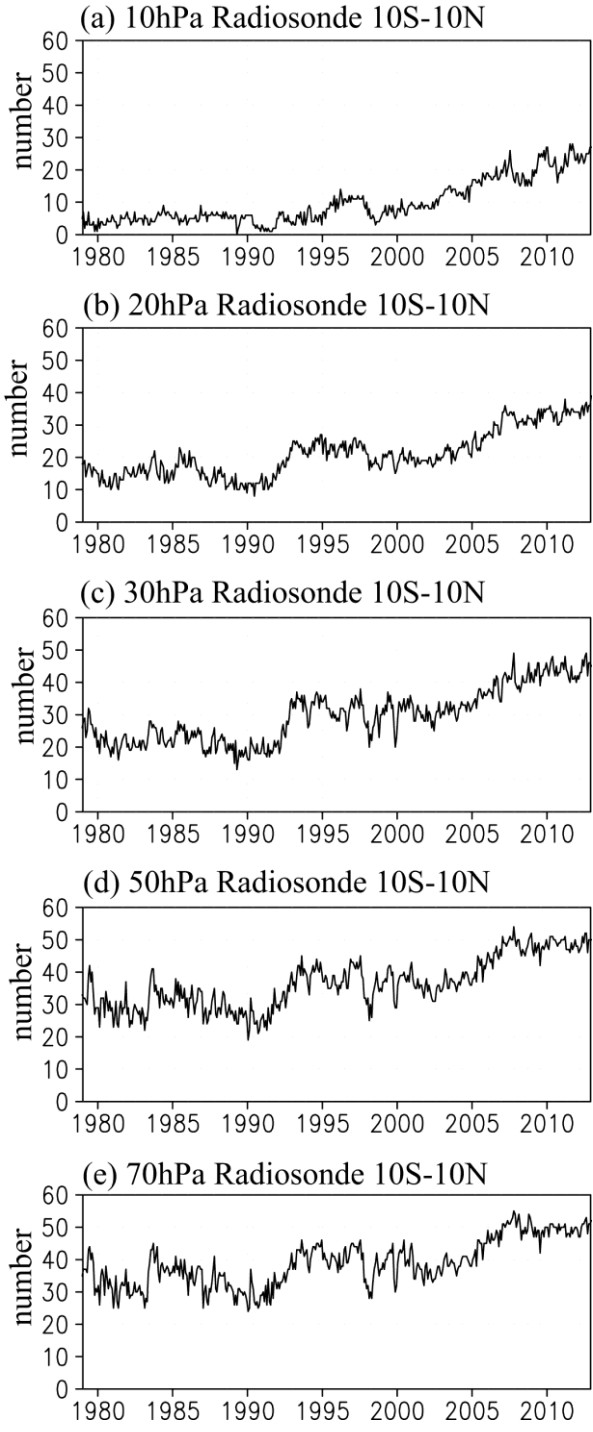

Figure 15. Time variation in number of radiosonde observations at 10°S–10°N at (a) 10 hPa, (b) 20 hPa, (c) 30 hPa, (d) 50 hPa, and (e) 70 hPa from 1979 to 2012.

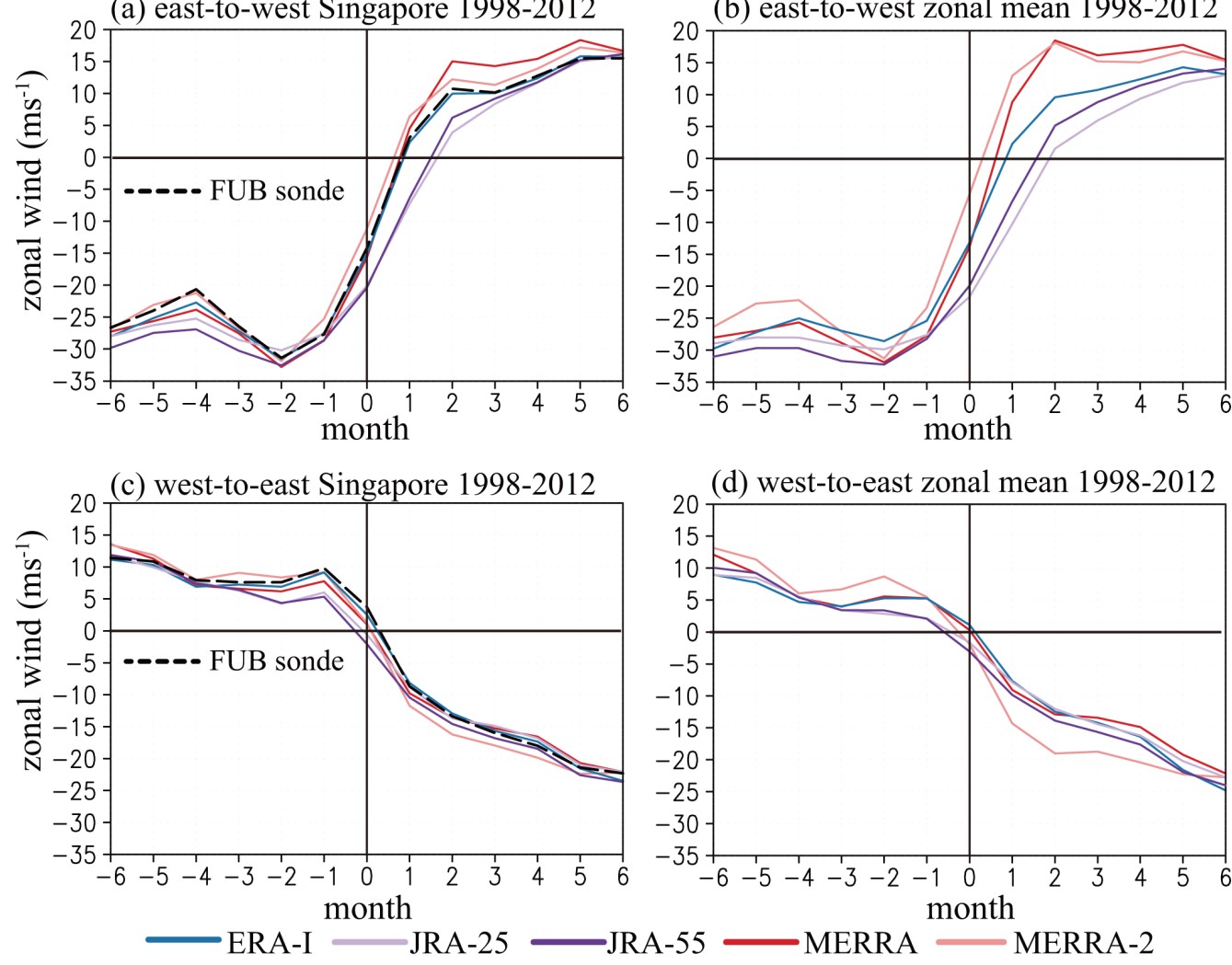

Figure 16. Same as Figs. 9a-d but during 1998 to 2012, excluding ERA-40 and including MERRA-2.

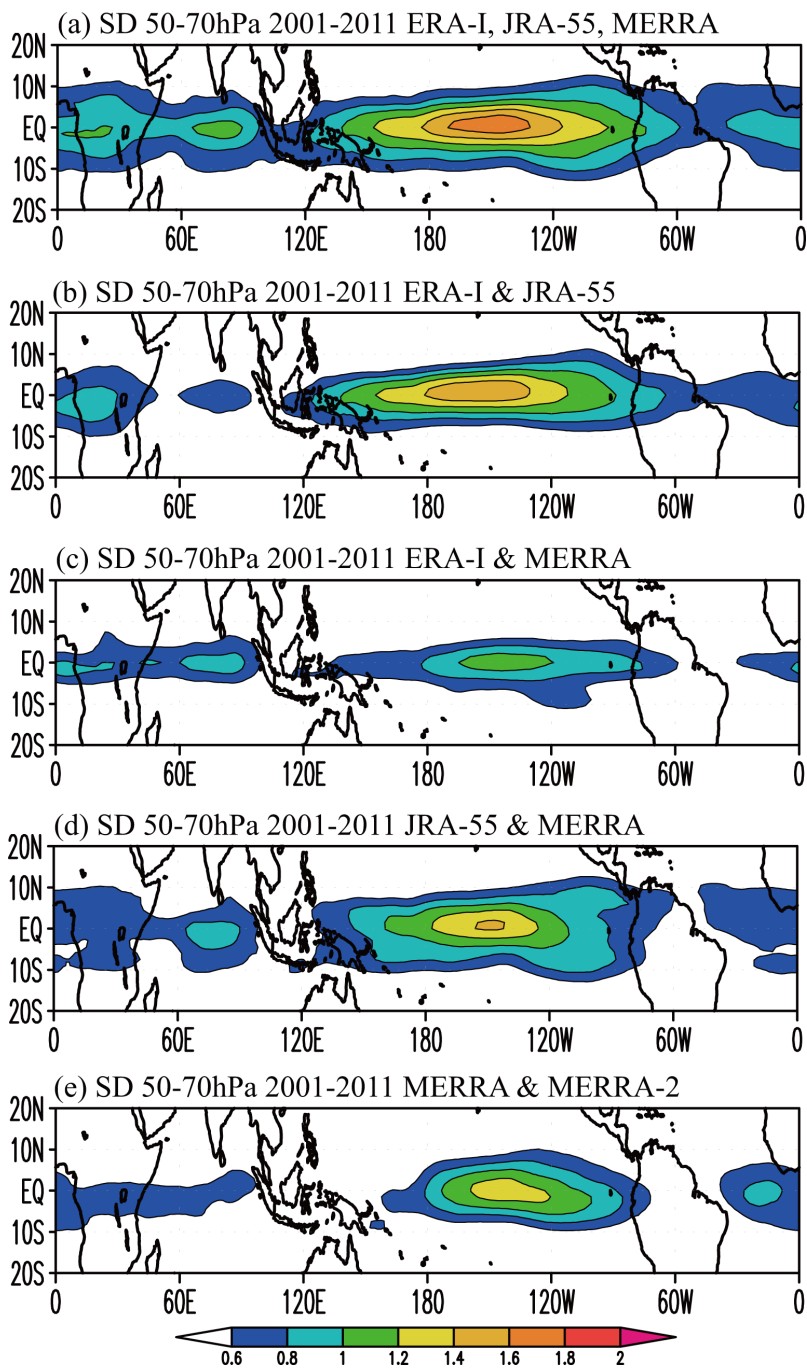

Figure 17. Same as Fig. 12c but for the standard deviation among (a) three reanalysis (ERA-I, JRA-55, and MERRA), (b) between ERA-I and JRA-55, (c) between ERA-I and MERRA, (d) between JRA-55 and MERRA, and (e) between MERRA and MERRA-2.