# Peer review of "Representation of the Tropical Stratospheric Zonal Wind in Global Atmospheric Reanalyses"

_Atmospheric Chemistry and Physics, 2016_

## Short Comment (SC1) · 23 Feb 2016

In this detailed study, *Kawatani et al.* (2016, hereinafter referred to as K16) present an intercomparison of monthly-mean zonal winds in the tropics in recent reanalyses. While focusing on monthly means is fully justified when tropical large-scale circulations (like the Quasi-Biennial Oscillation or quasi-stationnary planetary waves) are addressed, I would like to emphazise that such a time average ignores a significant fraction of the wind variability in the tropical lower stratosphere. Indeed, propagating dis-

turbances associated with planetary waves trapped in the equatorial wave guide (e.g., Kelvin and Rossby-gravity waves) are essentially discounted in K16, even though re-analyses in principle have sufficient horizontal and temporal resolution to resolve most of these waves. It might thus be appropriate that K16 briefly discuss the implications of such time averaging on their results. Two references are provided hereinunder to that purpose, and also to bring up some elements on reanalysis agreement vs accuracy.

1. K16 show that the agreement between reanalyses in monthly-mean zonal winds has continuously improved since 1979. The standard deviation (SD) among re-analyses reaches zonal-mean values of $\sim 1$ m s$^{-1}$ at 70 hPa in the last decade they studied (2001-2011), and never exceeds $1.8$ m s$^{-1}$ locally (their Figures 14 and 15). Despite the difficulties of constraining the tropical-stratosphere dynamics in atmospheric models with observations (which are recalled by K16), these SDs are quite impressive, as they are less than the assumed uncertainty associated with radiosonde winds in most models during the assimilation process. Yet, such good agreement likely does not apply to instantaneous reanalyses (i.e., without monthly average): *Baker et al.* (2014) (their Figure 2) have for instance shown that, in 2010, the zonal-mean SD between ECMWF operational analyses and NCEP GFS in zonal winds at 300 hPa is typically $3$ m s$^{-1}$, and can reach values over $5$ m s$^{-1}$ over the eastern Pacific and Indian Oceans, i.e. at least three times the values reported in K16: according to K16, the SD at 300 hPa should be less than in the lower stratosphere (their Figure 1 and 13).

2. Away from regions with assimilated observations, an agreement between re-analyses does not necessarily mean an equivalent agreement with observations, even in the most recent decade. For instance, *Podglajen et al.* (2014, hereinafter referred to as P14), who compared reanalyzed winds with independent in-situ observations performed along long-duration balloon flights in 2010, have reported occurrences where ECMWF operational analysis, ERA-interim and MERRA products all agree, while the balloon observations depart from them (see
their Figure 4). These events, which are associated with equatorial waves, induce discrepancies between reanalyses and observations that can reach values as large as $10 \text{ m s}^{-1}$ and last for weeks. They once again tend to occur over areas with (very) few radiosounding stations: the Eastern Pacific and Indian Ocean. In these areas, observational increments are very low (about $1 \text{ m s}^{-1}$ or less, see Figure 11 in P14), and the model dynamics is essentially running freely in the lower stratosphere.

It may therefore be worthwhile to warn the readers that they should not over-interpret encouraging figures regarding the improved agreement between reanalyses reported in K16.

I finally note that the lower agreement between reanalyses during QBO shear phases reported in K16 likely has a counterpart in the agreement between reanalyses and observations, as discussed in P14. Shear layers indeed tend to reduce the vertical wavelengths of waves that propagate in the shear direction while they increase the associated horizontal-wind disturbances. The reduced wavelength means that the model resolution may become insufficient to properly resolve the wave disturbances, even though the wave signal is present in the assimilated observations (see for instance Figure 9 in P14).

**References**

Baker, W. E., R. Atlas, C. Cardinali, A. Clement, G. Emmitt, B. M. Gentry, R. M. Hardesty, E. Källén, M. J. Kavaya, R. Langland, Z. Ma, M. Masutani, W. McCarty, R. B. Pierce, Z. Pu, L. P. Riishojgaard, J. Ryan, S. Tucker, M. Weissmann, and J. G. Yoe (2014), LIDAR-MEASURED WIND PROFILES The Missing Link in the Global Observing System, *Bull. Am. Meteorol. Soc.*, *95*, 543–564, doi:10.1175/BAMS-D-12-00164.1.

Kawatani, Y., K. Hamilton, K. Miyazaki, M. Fujiwara, and J. Anstey (2016), Representation of the tropical stratospheric zonal wind in global atmospheric reanalyses, *Atmos. Chem. Phys. Discuss.*, doi:10.5194/acp-2016-76.

Podglajen, A., A. Hertzog, R. Plougonven, and N. Žagar (2014), Assessment of the accuracy of (re)analyses in the equatorial lower stratosphere, *J. Geophys. Res.*, *119*, 11,166–11,188, doi:10.1002/2014JD021849.

---

## Referee Comment (RC1) · Anonymous Referee #1 · 1 Mar 2016

The authors evaluate the representation of the stratospheric equatorial zonal winds from nine different reanalysis products against each other and against radiosonde observations, in particular the FUB wind record. They focus largely on the inter-reanalysis standard deviation (which is generally largest in the deep tropics) and comparison with radiosonde observations as a means of evaluating the reanalyses.

In nearly all cases they find that this standard deviation is anti-correlated with the spatial and temporal availability of radiosonde observations. In the mid-stratosphere (10 hPa), the standard deviation is dominated by the zonal mean component which is largest during transitions of the QBO phase where the reanalyses tend to lag the observations by 2 weeks to 2 months, particularly in the easterly-to-westerly transition. The eddy component correlates with the QBO phase, being larger during the westerly phase, and is apparently associated with the representation of extratropical stationary waves. Lower

in the stratosphere and upper troposphere the eddy component becomes larger and the correlation with the QBO phase weakens. This structure appears to be associated with the stratospheric extension of the Walker circulation.

The analysis is a very useful contribution to the literature given the broad relevance of the QBO and the reliance of many studies on reanalysis products. The discussion is generally lucid and concise; my main criticism is that there are too many figures and it seems to me some of them could be removed (or combined) without impacting the central messages of the text. I would therefore recommend that the manuscript be accepted with minor revisions, either with a shortened figure list or stronger justifications in the text for those figures.

The discussion could also be strengthened by including some comments regarding the implications of these results for (a) studies using the reanalyses to understand the QBO or its impacts and (b) reanalysis centres trying to improve the representation of the winds.

With regards to (a), the bias in timings of the phase transitions could be relevant for studies which composite based on these dates, as could (possibly) the weak westerly wind maximum. The magnitude of the standard deviation in the horizontal winds throughout the tropics would also seem to be worth highlighting for trajectory studies given that there is some inclination to assume that all the inter-reanalysis differences are in the vertical velocities.

With regards to (b), one hypothesis that is raised is that having a forecast model with an internally generated QBO might reduce some of the biases. Given that MERRA 2 is in this category it seems that this hypothesis could be more explicitly tested; since it seems there are still significant biases, it would seem this is not sufficient to guarantee improved representation.

Could the errors (particularly in the mid-stratosphere) be associated with the slow advective propagation of information from the Singapore winds during phase transitions?

This would give you larger errors during transitions. Is it more likely the renalysis forecast models are systematically biasing the winds relative to radiosonde observations during transition periods?

Perhaps relevant to both (a) and (b), if reanalyses (or free running GCMs) are going to use existing winds to nudge towards a QBO, which reanalysis would the authors recommend (or perhaps another way to ask the question, are there any that should be cautioned against?)

Specific Comments:

Figure 1 and 2 contain much the same information, but Figure 2 is more useful; the former could be omitted without losing any conclusions. Also the latter could be improved if the labels and titles of each panel in 2 were removed (say, using a single labeled time axis and annotations within the panel) so that the lines were more easily seen.

Similarly Fig 6 contains the information in Fig. 5 and adds to it; Fig 5 can be omitted.

Fig 8: The time-dependence of the standard deviation would be much clearer if it was plotted on a different scale than the winds themselves. It also might be more informative to plot the standard deviations from each pressure level on the same axis so their temporal relationship can be more easily seen.

Fig. 9 and 18 could easily be combined.

Figs. 11 and 12 could also be combined and 11 (a-b) omitted.

p5 l21-25 It looks from Fig. 2d and e there are still significant anomalies in MERRA 2 at 50 hPa and 70 hPa. Are these really still likely to be be associated with an overactive SAO? This would seem to contradict the claim in l 26-27, though it is difficult to distinguish the MERRA and MERRA 2 curves.

p5 l29-30 What exactly was done with the tropical winds in NCEP-CFSR? Were they nudged towards ERA40 winds over this period? Is there a reference for this or will it be

mentioned somewhere in the SRIP report?

p6 l1-2 The justification for omitting MERRA 2 from much of the rest of the analysis is unclear to me, particularly since the authors return to it in Figs. 18 and 19. Why not just discuss it with the rest of the reanalyses? It also seems that NCEP-CFSR could be included for simplicity of method, though the case for omitting it is stronger. From Fig. 18 it looks like it does a good, if not better job of the transitions than other reanalyses; are the errors stronger in the middle of the QBO phase?

p6 l12-13 One of the main conclusions from Fig. 3 would seem to be that the SD is improving amongst more modern reanalyses products (at least up to MERRA - I'm guessing this would change if MERRA 2 were included here?)

p8 l10-15 Fig 13 a, b suggests that some of the structure in the eddy component in the tropical lower stratosphere might be associated with an extension of the Walker circulation - this is an interesting possibility and is distinct from issues of data availability. Since the phase of the QBO has not been considered in Fig. 13 would it make some sense to move the discussion on p 11 l 3-14 here? Also, as a test of this hypothesis, is the 70 hPa standard deviation correlated with the strength of the upper tropospheric Walker circulation?

p8 l24-26: Is there any correlation between the QBO phase and the number of radiosonde observations available?

p9 l10: This underestimation of the of the maximum westerly winds is one of the clearest biases and should be brought out more clearly in the conclusions (and possibly the abstract as well).

p9 21-25: This hypothesis could be evaluated explicitly here if the MERRA 2 winds were included. If they are not a good test of this hypothesis for some reason this could be explained here.

p10 l3-16: It's not clear to me why the authors have chosen to focus on a single case

here - surely more robust conclusions could be drawn by looking at the wavenumbers of the composited eddy component of the standard deviation? Indeed it might be interesting to see a zonal wave number spectrum of the standard deviation at several levels.

p12 l30: It would be interesting to test the importance of the satellite observations for the tropical winds by including JRA55c which only assimilates 'conventional' observations. This would seem to be a good way to strengthen many of the conclusions in this section, and is exactly the kind of question for which it is perfectly suited.

p23 l24-33: Given the close resemblance of the structures in Fig. 19 to other structures we've seen in many of the figures, I think these conclusions could be made without showing the figure explicitly.

---

## Referee Comment (RC2) · M.nbsp;A. Geller (Referee) · 3 Mar 2016

This is a very nicely written paper, but of course, I do have some suggestions for its improvement. I think this paper could benefit from a short discussion of the data assimilation process in the introduction. Such a discussion is not needed by those familiar with the data assimilation process, but many users of assimilation use the resulting products with out realizing that they are amalgams of data, the underlying model, and the statistical methods utilized. For this paper, I think it important at the beginning to indicate that the underlying model has its own climatology, and data, where present, nudges the resulting products toward observations. Also, unobserved quantities are adjusted to be consistent with the data being inserted together with the model climatology. To me, this together with the fact that as the Coriolis parameter tends toward zero the mass field constraint on the winds become weaker and weaker.

[Figure]

Another global comment is that reference should be made to Randel et al. (2004). Quoting from the summary section of that paper, "QBO variations in temperature and zonal wind are underestimated to some degree in most analyses, as compared to Singapore radiosonde data. The best results are derived from the assimilated datasets (ERA-40, ERA-15, METO, and NCEP, in that order) and only ERA-40 has realistic zonal wind amplitudes above 30 hPa. The use of balance winds in the Tropics (derived from geopotential data alone) is problematic for the QBO." The authors may wish to state to what extent they are updating those conclusions.

The following are some more detailed comments.

1. Page 2, line 8: Perhaps the paper by Yoo and Son might appear in time to be cited. It shows that the QBO exerts greater influence on the MJO than does ENSO. The reference is as follows: Yoo, C., and S.-W. Son, 2016: Modulation of the boreal wintertime Madden-Julian Oscillation by the stratospheric Quasi-Biennial Oscillation**, Geophysical Research Letters, accepted.

2. Page 2, line 14: The authors may wish to add a reference to Naujokat (1986) who states, ""The first three stations were used to produce a data set for the levels 70, 50, 40, 30, 20, 15, and 10 mb, which should be representative of the whole circumference at the equator since all investigations have shown that longitudinal differences in phase are small enough to be ignored." The implication here, not specifically said, is that QBO amplitude differences among the stations are more substantial, and likely cannot be ignored. This statement seems to be consistent with the conclusions in Hemilton et al. (2004). The authors might then go on to indicate whether they feel the reanalyses can capture such asymmetries. They might if the extropical planetary waves during QBO westerlies are well treated. Otherwise, I doubt they will.

3. Page 3 line 10: The situation is rapidly changing in that many models now produce spontaneous MJOs (i.e., GISS, CAM, etc.). Perhaps it would be better to say few GCMs used for reanalysis produce a spontaneous QBO.

4. Page 7, line 18: I believe it also depends on the climatology of the underlying GCM.

5. Page 10, lines 30-31: Do the authors have any idea why this might be so?

6. Page 14, lines 27-30: To what extent do the authors think this might affect the FUB QBO data set, which is often taken to represent the zonally averaged QBO?

Again, I want to emphasize that this is an excellent paper. The figures are excellent, and clearly indicate the points being made.

Marvin A. Geller

—————————————————————

---

## Referee Comment (RC3) · Anonymous Referee #3 · 10 Mar 2016

This paper assesses the tropical stratosphere variability in contemporary (and not so contemporary) climate reanalyses. Focus has been restricted to the effects of observational inhomogeneities going into the datasets, primarily from radiosondes, and how they constrain the observed features of the quasi-biennial oscillation and their potential role in the differences seen between datasets. It is found that the inter-model disagreement coincides to data poor areas, especially in the lower stratosphere. The reanalyses show good agreement over Singapore and show a progressive improvement at more recent times. Consistent and sizeable biases remain in the timing of the phases of the QBO especially during the easterly-westerly transitions at 10hPa.

This is a timely and well-written paper which will well-complement forthcoming science-focussed papers on tropical stratosphere variability. The number of figures is perhaps a little large, but they are generally of good quality. If the number of figure panels were

to be reduced, the ability to see fine features in the data would be really improved. I recommend publication pending due consideration of the points outlined below.

Main points:

I am unable to understand why it is difficult to establish what observations have gone into the various reanalyses. I would imagine the information is likely to be conspicuously posted on the individual Reanalyses Centres' websites or have been collated by other groups participating in the SPARC S-RIP project. I would have thought the Reanalysis Centres would find it particularly informative where (inter-model) differences are potentially coming from. This information should be included (or pointed to) in the paper, in some convenient way.

There is an inconsistent use of MERRA-2 data within the paper. As stated by the authors, MERRA-2 represents the ONE dataset whose forward model reproduces a QBO. This reviewer for one, would be especially keen to see that dataset assessed more throughout the paper. Is there a reason why 10hPa was chosen to assess the QBO phase timings? Outside of those times, there actually appears to be a better correspondence between MERRA-2 and the balloon record.

Conclusions referring to the wavenumber structure of the reanalyses' tropical wind need to be tempered a little. Figure 10 identifies a period where the high latitude stratosphere was particularly active. Other figures indicating wavenumber structures in longitude over a longer period of time refer to reanalyses differences (i.e. SD)

Finally, there are a lot of details going into the (numerous) figures. Some of these details were difficult to pick up even when blowing the figures up on screen. Can the authors make sure the figures are not lossy and try to improve the clarity between different models. The authors may also consider looking for a colour-blind-friendly contour scheme. Perhaps the authors should assess if they really need to include all 19 figures (>100 panels) in the main article.

[Figure]

Other points: (L28, P4) "...well-known summary..." (?)

(L9, P4) Presumably the interpolation to the FUB/IGRA data is done from the native reanalysis model resolution and not from the common ERA-I resolution mentioned in the previous sentence?

(L29, P5) Can the authors please find a suitable reference for the statement that NCEP-CFSR uses ERA40 winds in the tropics at and above 30hPA from 1 July 1981 to 31 December 1998. That is extraordinary!

(L10, P6) As the tropics are the focus for the paper, it would make sense to limit the latitudes in figure 3 to something like 20-30 degrees, for example. I do not think the extratropical SD differences show anything interesting anyway.

(L20, P6) Why have the authors chosen levels below 100hPa in figure 4? They should explain the reasoning behind this. Not much is written about this figure.

(L25, P6) Perhaps refer here to 'Indonesia' or the 'maritime continent' rather than the 'warm pool'. Also, the central (through to western) Pacific is where the most conspicuous SD values reside. This clarification may be important in pointing to other sources of model disagreement: in particular modes of (ocean-)atmosphere variability (e.g. El Nino-Modoki).

(L32, P7) Bogota data is on panel j not h. Also the authors should consider doing a an F-test to compare the differences in variances with and without IGRA data.

(L21, P9) It is a pity that MERRA-2 is not shown here, whilst perhaps attempting to minimise the strong SAO signal (at 10hPa) afflicting a couple of years early in the time record. As that model has a spontaneous QBO it would be very interesting to see the timing of the E/W and W/E transitions. It would clearly be less reliant on analysis increments and sufficient observations to constrain the QBO phase progression.

(L8, P10) It is evident that all 5 reanalyses shown do actually show a positive anomaly near the maritime continent, although there is evident a local maximum in 3-4 of the

datasets (probably highlighted due to contouring). Might also highlight (and reference) the fact a wave-1 warming occurred during December 1998 (and a wave-2 in February 1999)

(L17, P10) subtitle: "Dependence of the ? Difference..."

(L29, P10) "The overall larger SD in the westerly QBO phase, as compared to the easterly QBO phase,..."

(L7, P12) "...has reported stratospheric..."; reference, Pers. Comm.? The sentences following this need to be looked at. It is mentioned that a change in SD occurs around 1998, but then it is mentioned that there was a bias in the forward model of JRA-25. But presumably the bias in the forward model will not be responsible for changes around 1998 (forward model should not change during the reanalysis period - unlike operational analyses)

(L25..., p14) The authors should relax the statement referring to wavenumbers 1 and 2 dominating the 'eddy' zonal wind. Figure 10, mainly shows the zonal anomaly of zonal wind for January 1999 (a month sandwiched between two SSWs), so may not be representative of conditions at other times (e.g. 1979-2001)

---

## Referee Comment (RC4) · Anonymous Referee #4 · 17 Mar 2016

**General**

This paper is part of a series of papers reporting on the comparison of the representation of different aspects in major global atmospheric reanalysis datasets; here the focus is on the tropical stratospheric zonal wind and the QBO. The paper works out in detail the agreement and disagreement between the reanalyses. It also analyses the impact of different observations on the quality of the reanalysis. I recommend accepting the paper for publication in ACP, after some critique (see below) has been taken into account in the revised version.

The paper states that most free-running GCMs have problems simulating a realistic QBO. While I agree that the QBO is still a challenge for GCMs, progress has been made in recent years. Models that have addressed the major shortcoming of GCMs in this respect, namely the representation of atmospheric waves that contribute to driving the QBO, in particularly waves with short vertical wavelengths have shown success in allowing a quasi-biennial periodicity to emerge. Of course parametrised wave drag is still required to generate a realistic QBO. I suggest discussing these recent developments (Orr et al., 2010; Anstey et al., 2016) in a bit more detail.

Further, the QBO induces a secondary meridional circulation, i.e. QBO variations in meridional and vertical winds (Punge et al., 2009). As this point is both important for tropical transport and intimately related to the QBO, I suggest extending the analysis in the paper somewhat to cover this aspect. I think this could be an important contribution of this paper.

On many instances the paper points out observations of differences and aspects of the reanalyses that are interesting to note. For example the finding that quasistationary waves differ significantly among reanalyses. However, it is more important to make progress on finding the reasons for differences between reanalyses. Any suggestions in the paper how to make progress in this direction would be very helpful. I believe that it is beyond the scope of the paper to do assimilation experiments removing certain data sets from the analysis and in this way exploring the impact of particular data sets on the "message" of the reanalysis. But such studies have been done. And perhaps this paper could at least suggest ideas how to move forward in this direction in a discussion. For example what could be important and relevant assimilation experiments to perform?

In summary, I think the paper could be improved with respect to some aspects in the revision. I think it will ab a valuable contribution to ACP.

[Figure]

**Minor issues**

- p. 3, l 6: 'most such models'?

- p. 4, l. 5: there are also more recent publications on this point

- p 7., l 17: So is this only an expectation?

- p 7., l 32: off?

- p 9, l 4: if one writes \$-\$ in LATEX, then one obtains proper minus signs

- p 10, l 7: change 'represents' to 'shows'

- p 10, l. 26: continuous over which time period?

- p 11, l. 12: I agree it is interesting, but what is the conclusion here?

- p 11, l. 20: I agree it is interesting, but what is the conclusion here?

- p 11, l. 23: why is there a drop?

- p 12, l 5: How likely is the possible reason?

- p 12, l 7: change 'that' to 'on'

- p 12, l 8, 9: I think you mean the radiation code of the forecast model here.

- p 12, l 29: Discuss how this could be tested.

- p 13, l 14: the three most modern reanalyses

- p 14, l 4: why only 'nearly'?

- p 14, l 12: why?

- p 14, l 17, 18: It also means that the constraint is really necessary. So there are problems with the underlying model, correct?

- p. 16, Durre et al: abbreviate first names Fig 2, bottom panel: lines are difficult to disentangle

**References**

Anstey, J. A., Scinocca, J. F., and Keller, M.: Simulating the QBO in an atmospheric general circulation model: sensitivity to resolved and parameterized forcing, J. Atmos. Sci., 73, 1649–1665, doi: http://dx.doi.org/10.1175/JAS-D-15-0099.1, 2016.

Orr, A., Bechtold, P., Scinocca, J., Ern, M., and Janiskova, M.: Improved middle atmosphere climate and forecasts in the ECMWF model through a nonorographic gravity wave drag parameterization, J. Climate, 23, 5905–5926, doi: 10.1175/2010JCLI3490.1, 2010.

Punge, H. J., Konopka, P., Giorgetta, M. A., and Müller, R.: Effects of the quasi-biennial oscillation on low-latitude transport in the stratosphere derived from trajectory calculations, J. Geophys. Res., 114, D03102, doi: 10.1029/2008JD010518, 2009.

---

## Author Comment (AC1) · 10 May 2016

**Authors comments on "Representation of the Tropical Stratospheric Zonal Wind in Global Atmospheric Reanalyses" by Y. Kawatani, K. Hamilton, K. Miyazaki, M. Fujiwara, J. Anstey**

Corresponding author: Yoshio Kawatani (yoskawatani@jamstec.go.jp)

We are grateful to the four official referees for their helpful comments/suggestions and to Dr. Herzog for his contribution to the Open Discussion. We have revised the manuscript following their suggestions/comments. In this reply, we write the *reviewer's comments in blue italics*, while our responses are in regular fonts. At the end of this response are figures numbered R1-R6 which we prepared to help respond to the reviewers' comments, but do not propose to include in the revised manuscript.

Before presenting our detailed responses to each referee, let us address some overall issues that arose in some of the reviews.

Two of the four reviewers comment on the number of figures and panels in the manuscript. Reviewer #1 suggests some particular figures that could be removed or combined, while Reviewer #3 writes that "the number of figures is perhaps a little large" and suggests possibly reducing the total number of figure panels. We address the specifics in our responses to individual reviewers below where we explain that in many cases we prefer to keep the original figure. However, our efforts in this regard have resulted in a reduction of the 19 figures in the original version to 17 in our revision.

Some reviewers had suggestions for expanding the scope of our study in ways that would certainly be interesting, but which we feel are beyond the scope of our paper, which is strongly focused on what can be learned by intercomparing the monthly-mean zonal winds as represented in multiple state-of-the-art global reanalysis data sets. Notably Reviewer #4 suggests investigating the mean meridional circulation in the reanalyses noting the significance for chemical transport in the stratosphere. Abalos et al. (2015) have recently compared the stratospheric Brewer-Dobson circulation among MERRA, ERA-I and JRA55, while Miyazaki et al. (2015) did a somewhat similar study with 6 different reanalyses. Abalos et al. find quite large (~40%) differences among the reanalyses in the overall strength of the BD circulation and, specifically related to the QBO, they find that "There is a large spread among the estimates [of the fraction of variance in mean vertical motion explained by the QBO]". It is not clear that, even if we had the inclination to pursue this, we could add much to the Abalos et al. paper.

Extension of our study to include comparisons with special analyses that have been produced to test specific issues has been suggested. Reviewer 4 writes "I believe that it is beyond the scope of the paper to do assimilation experiments removing certain data sets from the analysis and in this way exploring the impact of particular data sets on the "message" of the reanalysis. But such studies have been done. And perhaps this paper could at least suggest ideas how to move forward in this direction" Along a similar line Reviewer 1 asks whether the JRA-55C reanalyses (which exclude all satellite data) could be included in our intercomparison. While we certainly agree that the comparison of analyses produced with subsets of the total input data can be illuminating, we feel this is beyond the scope of this paper and would to some extent duplicate the very recent study of Kobayashi et al. (2014) who directly compare the JRA-55C and JRA-55 reanalyses.

Reviewers #1 and #3 both raise concerns related to our treatment of the MERRA-2 reanalyses and wonder if it may be worthwhile to show more results directly comparing MERRA and MERRA-2. Since our original submission a very relevant new paper by Coy et al. (2016) has appeared in early

online release for J. Climate. Coy et al. (2016) is a detailed look at the QBO as represented in MERRA-2 and includes a number comparisons to MERRA. They provide some answers to the question of how the enhancement in the gravity wave parameterization employed in the dynamical model for MERRA-2 affects the representation of the QBO. Coy et al find that "the increased equatorial gravity wave drag in MERRA-2 has reduced the zonal wind data analysis contribution compared to MERRA…"  However Coy et al. also note the issues with fairly pronounced deficiencies in the early period covered by MERRA-2, specifically noting that before 1995 "MERRA-2 appears to overemphasize the annual cycle". This makes it problematic for us to include MERRA-2 in our calculation of the SD among all the other analyses over the long periods we considered. In our revision we have retained our basic approach which is to exclude MERRA-2 from our calculations except when showing the wind evolution during the rapid transition periods for individual reanalyses (Fig. 16 of revised version) composited just for the post-1998 period, where the comparison with MERRA is particularly relevant. Our conclusions on the likely role of the gravity wave drag in MERRA-2 seem to be consistent with the findings of Coy et al.

**Reply to anonymous referee #1**

*The authors evaluate the representation of the stratospheric equatorial zonal winds from nine different reanalysis products against each other and against radiosonde observations, in particular the FUB wind record. They focus largely on the inter-reanalysis standard deviation (which is generally largest in the deep tropics) and comparison with radiosonde observations as a means of evaluating the reanalyses.*

*In nearly all cases they find that this standard deviation is anti-correlated with the spatial and temporal availability of radiosonde observations. In the mid-stratosphere (10 hPa), the standard deviation is dominated by the zonal mean component which is largest during transitions of the QBO phase where the reanalyses tend to lag the observations by 2 weeks to 2 months, particularly in the easterly-to-westerly transition. The eddy component correlates with the QBO phase, being larger during the westerly phase, and is apparently associated with the representation of extratropical stationary waves. Lower in the stratosphere and upper troposphere the eddy component becomes larger and the correlation with the QBO phase weakens. This structure appears to be associated with the stratospheric extension of the Walker circulation.*

*The analysis is a very useful contribution to the literature given the broad relevance of the QBO and the reliance of many studies on reanalysis products. The discussion is generally lucid and concise; my main criticism is that there are too many figures and it seems to me some of them could be removed (or combined) without impacting the central messages of the text. I would therefore recommend that the manuscript be accepted with minor revisions, either with a shortened figure list or stronger justifications in the text for those figures.*

Thank you very much for your suggestion. As noted above we have tried to reduce/combine figures and have succeeded in reducing the total number of figures to 17 from the 19 in the original version of the manuscript.

*The discussion could also be strengthened by including some comments regarding the implications of these results for (a) studies using the reanalyses to understand the QBO or its impacts and (b) reanalysis centres trying to improve the representation of the winds.*

*With regards to (a), the bias in timings of the phase transitions could be relevant for studies which*

*composite based on these dates, as could (possibly) the weak westerly wind maximum. The magnitude of the standard deviation in the horizontal winds throughout the tropics would also seem to be worth highlighting for trajectory studies given that there is some inclination to assume that all the inter-reanalysis differences are in the vertical velocities.*

These are useful points to emphasize and in the revision we have added these to the discussion in the explanation of Figs. 9 and 16 and in the summary session.

*With regards to (b), one hypothesis that is raised is that having a forecast model with an internally generated QBO might reduce some of the biases. Given that MERRA 2 is in this category it seems that this hypothesis could be more explicitly tested; since it seems there are still significant biases, it would seem this is not sufficient to guarantee improved representation.*

As we noted above Coy et al. (2016) have very recently examined this issue in some detail. They found that MERRA-2 has reduced the zonal wind analysis increments compared to MERRA, so that the QBO mean meridional circulation can be expected to be more physically forced and more physically consistent. In our revision we include these findings of Coy et al. in the discussion section.

*Could the errors (particularly in the mid-stratosphere) be associated with the slow advective propagation of information from the Singapore winds during phase transitions?*

*This would give you larger errors during transitions. Is it more likely the reanalysis forecast models are systematically biasing the winds relative to radiosonde observations during transition periods?*

These are two good suggestions that may help explain the lag in the reanalysis winds during the QBO transitions. We had already included the second possibility in our original manuscript, i.e. we had speculated that the delay could be understood as a consequence of the dynamical model bias (see discussion starting at page 9, line 21, in the original manuscript). In our revision we include also the possibility that during periods of weak zonal wind there will be delays in the zonal advective propagation of information introduced into the analysis system from observations at individual stations (notably Singapore). We are not able to definitely assess these two possible mechanisms, but the fact that the lag in the reanalysis winds is more pronounced in the more rapid easterly-to-westerly wind transition may favor the importance of the model bias over the slow advective propagation mechanism.

*Perhaps relevant to both (a) and (b), if reanalyses (or free running GCMs) are going to use existing winds to nudge towards a QBO, which reanalysis would the authors recommend (or perhaps another way to ask the question, are there any that should be cautioned against?)*

Our results support two strong conclusions in this regard: that the NCEP reanalyses are notably deficient in their representation of the QBO, and that systematic anomalies in the annual cycle contaminate the first part of the MERRA-2 record (a point now confirmed by Coy et al., 2016). Beyond that we have not tried to identify overall "best" analyses and have no results that strongly favor or disfavor application of individual data sets. However if the metric is how well the monthly mean zonal winds in the reanalyses compare with high quality direct station balloon observations, then ERA-I overall has a slight edge over the other data sets, as shown in Figs. 2, 9, 18 in the original manuscript (Figs. 2, 9, 16 in the revised manuscript). We add a comment "The RMS differences from FUB values in 1979–2012 are smallest in ERA-I, while those in NCEP-1 and NCEP-2 are much larger than those in the other reanalyses" in the discussion of Fig. 2f.

*Specific Comments:*
*Figure 1 and 2 contain much the same information, but Figure 2 is more useful; the former could be omitted without losing any conclusions. Also the latter could be improved if the labels and titles of each panel in 2 were removed (say, using a single labeled time axis and annotations within the panel) so that the lines were more easily seen. Similarly Fig 6 contains the information in Fig. 5 and adds to it; Fig 5 can be omitted.*

We considered the reviewer's suggestions here and also tried to reconsider the value of each of the figures in the original version. We decided to remove the original Fig. 4 (showing SD among reanalyses data sets as a function of height) as perhaps the least interesting to readers. The reviewer is correct that Figs. 1 and 2 present the same information. However, we decided to retain Fig.1 (simple height-time sections of zonal mean equatorial zonal-mean zonal wind) as the most basic presentation of the data sets considered. In principle, anyone could trivially duplicate this by simply grabbing publically available data sets and making contour plots, but we feel many readers will appreciate the considerable effort we put in to actually produce a single legible figure summarizing these data conveniently. The reviewer is also correct that the information in the shading of Fig. 5 is reproduced in the contours of Fig. 6. We arranged this so that the reader's attention could be first drawn to the overall structure of the SD in Fig. 5 without the complication of the station locations. We feel this allows for a logical exposition and sets up the reader to appreciate the more involved comparison in Fig. 6.

*Fig 8: The time-dependence of the standard deviation would be much clearer if it was plotted on a different scale than the winds themselves. It also might be more informative to plot the standard deviations from each pressure level on the same axis so their temporal relationship can be more easily seen.*

This is an excellent point. We changed the figure in accord with this suggestion and it is definitely clearer now.

*Fig. 9 and 18 could easily be combined.*

Indeed the panels from these two figures could be combined. However, given the discussion surrounding the original Fig. 18 (Fig. 16 of the revision) comes quite a bit later in the paper and concerns only the post-1998 data, we feel it is easier for the reader if we retain the figures with separate numbers.

*Figs. 11 and 12 could also be combined and 11 (a-b) omitted.*

We combined the original Figs. 11 and 12, but felt that 11a,b was useful for the reader and retained these panels as the a,b of the new figure (Fig. 11 in the revision).

*p5 l21-25 It looks from Fig. 2d and e there are still significant anomalies in MERRA2 at 50 hPa and 70 hPa. Are these really still likely to be associated with an overactive SAO? This would seem to contradict the claim in l 26-27, though it is difficult to distinguish the MERRA and MERRA 2 curves.*

As noted earlier Coy et al. (2016) addressed this and indicated that the deficiencies in the early period covered by MERRA-2 appear to be related to representation of the annual cycle. At 50hPa, the MERRA‑2 winds are not so anomalous, however. It is the NCEP reanalyses that have larger apparent anomalies at 50 hPa.

*p5 l29-30 What exactly was done with the tropical winds in NCEP-CFSR? Were they nudged towards ERA40 winds over this period? Is there a reference for this or will it be mentioned somewhere in the SRIP report?*

Saha et al. 2010 mentioned in their section "QBO PROBLEM IN THE GSI" that "In order that the streams could proceed with a reasonable QBO signature, it was decided that the ERA-40 stratospheric wind profiles should be used as bogus observations for the period from 1 July 1981 to 31 December 1998". We add this reference here.

*p6 l1-2 The justification for omitting MERRA 2 from much of the rest of the analysis is unclear to me, particularly since the authors return to it in Figs. 18 and 19. Why not just discuss it with the rest of the reanalyses? It also seems that NCEP-CFSR could be included for simplicity of method, though the case for omitting it is stronger. From Fig. 18 it looks like it does a good, if not better job of the transitions than other reanalyses; are the errors stronger in the middle of the QBO phase?*

As noted earlier, we identified a peculiar anomaly in the representation of the annual cycle in the first part of the MERRA-2 record, a point now confirmed by Coy et al. (2016). This makes us reluctant to include MERRA-2 in our main results, namely calculations of SD among several analyses over long periods. However, given the interest in the accuracy of the reanalyses during the rapid transition phases, we did bring MERRA-2 into the discussion surrounding Fig 18 (Fig 16 in the revision), because the fact that the MERRA-2 dynamical model produces a QBO seems particularly relevant here. To avoid the problems with MERRA-2, of course, we limited the period considered in this one case to just post 1997. If we add MERRA-2 results to the original Fig. 9 (and adjust for the fact that MERRA-2 only starts in 1980) then we get the Fig. R6 below, where the MERRA-2 results really look peculiar.

With regard to NCEP-CSFR, once again the fact that these analyses were constrained to very nearly agree with ERA-40 above 30 hPa makes their inclusion into our calculations of SD among several analyses rather problematic.

*p6 l12-13 One of the main conclusions from Fig. 3 would seem to be that the SD is improving amongst more modern reanalyses products (at least up to MERRA - I'm guessing this would change if MERRA 2 were included here?)*

Indeed adding MERRA-2 makes the overall results worse (higher SD). This can be verified in Fig. R1 below, where we show how Fig. 3 changes when we include MERRA-2 (we have had to restrict the period to 1980-2012, as MERRA-2 does not begin until 1980).

*p8 l10-15 Fig 13 a, b suggests that some of the structure in the eddy component in the tropical lower stratosphere might be associated with an extension of the Walker circulation - this is an interesting possibility and is distinct from issues of data availability. Since the phase of the QBO has not been considered in Fig. 13 would it make some sense to move the discussion on p 11 l 3-14 here? Also, as a test of this hypothesis, is the 70 hPa standard deviation correlated with the strength of the upper tropospheric Walker circulation?*

In our revised version we have moved the discussion of Fig. 13 (Fig. 7 in the revision) to the last part of Section 3.2, as suggested.

Fig. R2 below shows the longitude-height cross section of the temporal correlation for the period

1979-2001 between the absolute value of the zonal wind (i.e., the strength of the zonal wind) and the the standard deviation among reanalyses averaged over 10ºN-10ºS (i.e., temporal correlation for which the two time series are |[u]| and SD calculated following Eq. 1).

In the upper part of the Walker circulation, the relatively high positive correlation is seen in the eastern hemisphere, while the correlation is relatively low in the central Pacific. In the mean state, the eddy component in the tropical lower stratosphere might be associated with an extension of the Walker circulation, but 70hPa standard deviation is relatively small (Fig.13b in the original manuscript), and it seems not correlated with the strength of the upper tropospheric Walker circulation. The large SD from the upper troposphere to the stratosphere in the central Pacific could be simply related to the fewer in-situ observations available there. In the middle stratosphere, the correlation is negative, corresponding to large SD during the phase transition of the QBO. We add this figure in Fig.7c in the revised manuscript.

*p8 l24-26: Is there any correlation between the QBO phase and the number of radiosonde observations available?*

We calculated the correlation between the number of radiosonde observations (Fig. 17 in the original manuscript) and FUB zonal wind at each height, and confirmed that there is no correlation between the QBO phase and observational numbers at all heights of 10 to 70 hPa.

*p9 l10: This underestimation of the maximum westerly winds is one of the clearest biases and should be brought out more clearly in the conclusions (and possibly the abstract as well).*

This issue is a little subtle, as the statement we made at this point in the manuscript applies to the maximum winds in a composite stretching over a finite period (6 months) from the transition date. This is not quite the same as a simple statement that the QBO westerly extremes are underestimated in the reanalyses, so we are reluctant to call this point out in the Abstract and Conclusion.

*p9 21-25: This hypothesis could be evaluated explicitly here if the MERRA 2 winds were included. If they are not a good test of this hypothesis for some reason this could be explained here.*

When we include MERRA-2 in the original Fig. 9 (and adjust for the fact that MERRA-2 only starts in 1980) then we get the Fig. R6 below, where the MERRA-2 results really look peculiar. We return to this issue in the discussion around Figure 18a (Fig. 16a of the revised version) that includes the later part of the MERRA-2 data record. This shows that the MERRA-2 reanalyses display an easterly-to-westerly phase transition at Singapore that is even more rapid than in the direct balloon observations. These results may indicate that the gravity wave sources in MERRA-2 are now excessive. We have extended our discussion here in the revised manuscript in the discussion of Fig. 16.

*p10 l3-16: It's not clear to me why the authors have chosen to focus on a single case here - surely more robust conclusions could be drawn by looking at the wavenumbers of the composited eddy component of the standard deviation? Indeed it might be interesting to see a zonal wave number spectrum of the standard deviation at several levels.*

The point is that the single case we showed is quite typical of the quasi-stationary wave behavior throughout the record. Fig. R3 shows longitudinal variations of the *u'* (deviation of the zonal wind values from the zonal mean) at 10hPa over the equator in each January from 1990 to 2000. Please note that Fig.10g in the original manuscript showed zonal wind *u* in 1996 and 1999 in the same panel,

but here we show *u'* at each year separately.

Figs. R4 and R5 show *u'* at 10hPa in each of five reanalyses in those Januaries when the equatorial zonal mean zonal wind is westerly (i.e., $[\bar{u}]_{eq} > 0$) in 1992, 1993, 1995 and 1997.

So we have checked other years and found the monthly mean eddy structures in the middle stratosphere associated with quasi-stationary planetary waves are qualitatively similar among years. We explain this more clearly in the revised version of the paper in the discussion of Fig. 10.

*p12 l30: It would be interesting to test the importance of the satellite observations for the tropical winds by including JRA55c which only assimilates 'conventional' observations. This would seem to be a good way to strengthen many of the conclusions in this section, and is exactly the kind of question for which it is perfectly suited.*

We agree that insights can be found through comparisons of full reanalyses with versions that have had some data inputs withheld, and indeed the JRA-55C data provide such a possible comparison. As noted earlier, however, we think that adding such comparisons is beyond the scope of the present paper which is strongly focused on what can be learned by intercomparing the monthly-mean zonal winds as represented in multiple state-of-the-art global reanalysis data sets. We note also that the recent study of Kobayashi et al. (2014) comparing the JRA-55C and JRA-55 reanalyses presents results related to the stratospheric QBO in summary section.

*p23 l24-33: Given the close resemblance of the structures in Fig. 19 to other structures we've seen in many of the figures, I think these conclusions could be made without showing the figure explicitly.*

We feel that showing a measure of how well the most "up-to-date" reanalyses perform in the most recent period will be of interest to many readers, and so in our revision we retained Fig.19 (Fig. 17 in the revised version).

**Reply to referee #2, Prof. Marvin A. Geller**

*This is a very nicely written paper, but of course, I do have some suggestions for its improvement. I think this paper could benefit from a short discussion of the data assimilation process in the introduction. Such a discussion is not needed by those familiar with the data assimilation process, but many users of assimilation use the resulting products without realizing that they are amalgams of data, the underlying model, and the statistical methods utilized. For this paper, I think it important at the beginning to indicate that the underlying model has its own climatology, and data, where present, nudges the resulting products toward observations. Also, unobserved quantities are adjusted to be consistent with the data being inserted together with the model climatology. To me, this together with the fact that as the Coriolis parameter tends toward zero the mass field constraint on the winds become weaker and weaker*

Thank you very much for your suggestions. The core members of S-RIP are now preparing the detailed introduction of the reanalysis data, but this paper will not be available before this study will be published. So, in our revised version we have included a short discussion of the data assimilation process in the introduction as follows.

"Data assimilation is the technique for combining different observational data sets with a model, by considering the characteristics of each measurement and taking into account errors in both the

measurements and the model (e.g., Kalnay, 2003). Advanced data assimilation schemes like the 4D-Var technique use the information provided by various measurements, such as radiosonde and satellite-derived measurements, and propagates it, in time and space, from a limited number of observable variables to a wide range of meteorological variables to provide global fields that are dynamically consistent and in agreement with the observations. Meteorological reanalyses have been conducted at operational centers using various approaches, which ingest a variety of observations over the period of each reanalysis product. Differences in the forecast model, assimilated measurements, and data assimilation technique used for producing reanalysis datasets can lead to differences in their representation of the mean state, variability, and long-term trend of atmospheric fields"

*Another global comment is that reference should be made to Randel et al. (2004). Quoting from the summary section of that paper, "QBO variations in temperature and zonal wind are underestimated to some degree in most analyses, as compared to Singapore radiosonde data. The best results are derived from the assimilated datasets (ERA-40, ERA-15, METO, and NCEP, in that order) and only ERA-40 has realistic zonal wind amplitudes above 30 hPa. The use of balance winds in the Tropics (derived from geopotential data alone) is problematic for the QBO." The authors may wish to state to what extent they are updating those conclusions.*

We have added the reference to Randel et al. (2004). We agree the use of balance winds in the tropics is problematic for the QBO and this largely explains why the SD peaks so strongly on the equator (less constraint on the winds from satellite temperature retrievals at low latitudes). Our brief review of the history noted that ERA-40 is an improvement over ERA-15 and our own results showed deficiencies in the NCEP reanalyses. So our results seem consistent with the conclusions of Randel et al. (2004). Now, other newer reanalyses are available. MERRA-2 is unique because this model can simulate the QBO internally but has some other problems (see Coy et al., 2016, and also our discussion at several other points in this Authors' Reply).

*The following are some more detailed comments.*
*1. Page 2, line 8: Perhaps the paper by Yoo and Son might appear in time to be cited. It shows that the QBO exerts greater influence on the MJO than does ENSO. The reference is as follows: Yoo, C., and S.-W. Son, 2016: Modulation of the boreal wintertime Madden-Julian Oscillation by the stratospheric Quasi-Biennial Oscillation, Geophysical Research Letters, accepted.*

We include the reference of Yoo and Son (2016) in the introduction of our revised manuscript.

*2. Page 2, line 14: The authors may wish to add a reference to Naujokat (1986) who states, ""The first three stations were used to produce a data set for the levels 70, 50, 40, 30, 20, 15, and 10 mb, which should be representative of the whole circumference at the equator since all investigations have shown that longitudinal differences in phase are small enough to be ignored." The implication here, not specifically said, is that QBO amplitude differences among the stations are more substantial, and likely cannot be ignored. This statement seems to be consistent with the conclusions in Hamilton et al. (2004). The authors might then go on to indicate whether they feel the reanalyses can capture such asymmetries. They might if the extropical planetery waves during QBO westerlies are well treated. Otherwise, I doubt they will.*

Indeed our current results are in general agreement with those Hamilton et al. (2004). Specifically, like Hamilton et al. we find modest, but significant, systematic zonal asymmetries in the QBO. We agree with Dr. Geller that this has implications for the application of the FUB "single station" record as it splices data from three separate stations during different epochs. In the present paper all the data

we employ is from the post-1979 period when the FUB data were entirely from Singapore. As to the question of whether the planetary waves are well treated in the reanalyses, our contribution in this paper is to intercompare the reanalyses. We find an overall basic agreement among reanalyses, but considerable variation in the exact structure and amplitude of the zonal asymmetries (see the analysis of the SD presented in the manuscript and also Figs. R4 and R5 below). Here we explain as follows: "The high quality of these balloon data, and the close proximity of the stations to the equator, has led FUB series to be widely used, despite being based on only a single station each month (and despite modest inhomogeneities that the changes of station location may introduce into the record; see Section 3 below)"

*3. Page 3 line 10: The situation is rapidly changing in that many models now produce spontaneous MJOs (i.e., GISS, CAM, etc.). Perhaps it would be better to say few GCMs used for reanalysis produce a spontaneous QBO.*

We believe Dr. Geller means "QBOs" here, not "MJOs". In our revised version we have been more explicit and said that "because most GCMs display fairly steady, weak prevailing zonal winds due to failure to reproduce a spontaneous QBO".

*4. Page 7, line 18: I believe it also depends on the climatology of the underlying GCM.*

In our revised version we have added "In addition, the SD may also reflect differences among the climatologies of the GCMs used in the analysis systems" here.

*5. Page 10, lines 30-31: Do the authors have any idea why this might be so?*

The main point seems to be that the quasi-stationary waves propagating in from the NH extratropics are quite weak below 30 hPa, a result that is consistent with the earlier findings of Hamilton et al. (2004).

*6. Page 14, lines 27-30: To what extent do the authors think this might affect the FUB QBO data set, which is often taken to represent the zonally averaged QBO?*

Following our comments above, indeed the systematic zonal asymmetry in the zonal wind QBO should introduce discontinuities in the FUB data set which is spliced together from observational records at Canton Island, Gan and Singapore. However, as discussed in Hamilton et al. (2004), these stations are confined zonally to 73E-172W and the contrasts in the QBO near the equator within this sector are fairly small (see Hamilton et al.).

*Again, I want to emphasize that this is an excellent paper. The figures are excellent, and clearly indicate the points being made.*

We appreciate your evaluation and encouragement.

**Reply to anonymous referee #3**

*This paper assesses the tropical stratosphere variability in contemporary (and not so contemporary) climate reanalyses. Focus has been restricted to the effects of observational inhomogeneities going into the datasets, primarily from radiosondes, and how they constrain the observed features of the quasi-biennial oscillation and their potential role in the differences seen between datasets. It is found*

*that the inter-model disagreement coincides to data poor areas, especially in the lower stratosphere. The reanalyses show good agreement over Singapore and show a progressive improvement at more recent times. Consistent and sizeable biases remain in the timing of the phases of the QBO especially during the easterly-westerly transitions at 10hPa.*

*This is a timely and well-written paper which will well-complement forthcoming science-focussed papers on tropical stratosphere variability. The number of figures is perhaps a little large, but they are generally of good quality. If the number of figure panels were to be reduced, the ability to see fine features in the data would be really improved. I recommend publication pending due consideration of the points outlined below.*

As described in our response too Reviewer #1, we have tried to reconsider the value of each of our figures and decided to remove Fig. 4 and combine the panels in Figs. 11 and 12 (to produce Fig. 11 of the revised version).

*Main points:*
*I am unable to understand why it is difficult to establish what observations have gone into the various reanalyses. I would imagine the information is likely to be conspicuously posted on the individual Reanalyses Centres' websites or have been collated by other groups participating in the SPARC S-RIP project. I would have thought the Reanalysis Centres would find it particularly informative where (inter-model) differences are potentially coming from. This information should be included (or pointed to) in the paper, in some convenient way.*

We have contacted professionals in each reanalysis center and several members of S-RIP, and confirmed that the situation is as follows. Among the reanalysis data sets considered in this paper, only MERRA provides day-to-day observation information including detailed geolocations in a gridded form. Making such information publicly available in a user friendly manner was not feasible due to the volume and the complexity of observation data. Friendly formats and tools are currently being developed or considered at some reanalysis centers for their future reanalysis products. Furthermore, differences between different reanalyses are not just related to which observations were actually used (though this might have the largest impact), but might also depend on how a given data set was used, e.g., what data-quality-control and bias-correction procedures were actually applied or not applied. These procedures are different in the details for different reanalyses.

Currently, two manuscripts are being prepared. One is the S-RIP 2016 interim Report where there is a chapter on "Description of the Reanalysis Systems" written by Wright et al., and the other is an ACP overview paper on the special issue on "The SPARC Reanalysis Intercomparison Project (S-RIP)" by Fujiwara, Wright et al. Researchers from all the reanalysis centers are also coauthors of these manuscripts, having provided very comprehensive depiction of all the reanalyses including the information not given in the reanalysis reference papers. Please see these manuscripts once they become available.

*There is an inconsistent use of MERRA-2 data within the paper. As stated by the authors, MERRA-2 represents the ONE dataset whose forward model reproduces a QBO. This reviewer for one, would be especially keen to see that dataset assessed more throughout the paper.*

As we discussed earlier, the inclusion of MERRA-2 is problematic for our main calculations which relate to the differences among several reanalyses over long periods. Since our original submission a very relevant new paper by Coy et al. (2016) has appeared in early online release for *J. Climate*. Coy et al. (2016) is a detailed look at the QBO as represented in MERRA-2 and includes a number

comparisons to MERRA. However Coy et al. specifically call out the issues with fairly pronounced deficiencies in the early period covered by MERRA-2, noting that before 1995 "MERRA-2 appears to overemphasize the annual cycle". This makes it problematic for us to include MERRA-2 in our calculation of the SD among all the other analyses over the long periods we considered. In our revision we have retained our basic approach which is to exclude MERRA-2 from our calculations except when showing the wind evolution during the rapid transition periods for individual reanalyses (Fig. 16 of revised version, composited just for the post-1998 period) where the comparison with MERRA is particularly relevant. Some measure of the difficulty in including the earlier MERRA-2 data is apparent in the Fig. R6 where we have included added MERRA-2 to the transition composites computed over a long period.

*Is there a reason why 10hPa was chosen to assess the QBO phase timings? Outside of those times, there actually appears to be a better correspondence between MERRA-2 and the balloon record.*

We chose 10 hPa as presenting the biggest challenge for the reanalyses as the numbers of radiosonde observations are fewest and the SD among reanalyses is largest. We add this sentence in the explanation of Fig.9.

*Conclusions referring to the wavenumber structure of the reanalyses' tropical wind need to be tempered a little. Figure 10 identifies a period where the high latitude stratosphere was particularly active. Other figures indicating wavenumber structures in longitude over a longer period of time refer to reanalyses differences (i.e. SD)*

We cannot include all years considered but show one particular year. We did same analysis in other years and confirmed the structures of quasi-stationary planetary waves are similar in spite of different amplitude. We have confirmed that the characteristics, namely (i) the eddy component near the equator appears to be dominated by zonal wavenumber 1 and 2 quasi-stationary planetary waves propagating from mid-latitudes during westerly phase of the QBO, and (ii) the eddy components are very small over the equator in the easterly phase of the QBO, are qualitatively similar in different years. The fact that eddy components are large during westerly phase of the QBO is shown in Fig. 9f for 1979-2001. In our revision we have added a sentence here: "these characteristics seen in westerly and easterly phase of the QBO are qualitatively similar among other years".

*Finally, there are a lot of details going into the (numerous) figures. Some of these details were difficult to pick up even when blowing the figures up on screen. Can the authors make sure the figures are not lossy and try to improve the clarity between different models. The authors may also consider looking for a colour-blind-friendly contour scheme. Perhaps the authors should assess if they really need to include all 19 figures (>100 panels) in the main article.*

Thank you for these comments. We are not sure exactly which figures the reviewer is concerned about here. The most problematic may be Fig. 7 (Fig. 6 in the revised manuscript) which was assembled in Illustrator from many individual panels. Even when this figure is blown up to 400% (so one panel spans the whole viewable space on the screen), the curves and lettering are still reasonably crisp. For this figure we feel that it is useful for the reader to see all the panels together on one page, so we are not inclined to try to address any possible issue of crispness by making this (or other figures) span multiple pages. We feel the legibility of the figures is not a serious problem. We had not considered the issue of whether our color schemes are best for color blind readers. Perhaps in the near future this issue could be taken up by the journal publishers or the research community at large, and suitable guidance issued for authors.

*Other points: (L28, P4) "...well-known summary..." (?)*

In the revision we have removed this characterization.

*(L9, P5) Presumably the interpolation to the FUB/IGRA data is done from the native reanalysis model resolution and not from the common ERA-I resolution mentioned in the previous sentence?*

For the results presented we did interpolate from the common ERA-I resolution. There is no significant difference if we interpolate from the native reanalysis model resolutions.

*(L29, P5) Can the authors please find a suitable reference for the statement that NCEPCFSR uses ERA40 winds in the tropics at and above 30hPA from 1 July 1981 to 31 December 1998. That is extraordinary!*

We refer to Saha et al. 2010 here, who stated in their section "QBO PROBLEM IN THE GSI" that "In order that the streams could proceed with a reasonable QBO signature, it was decided that the ERA-40 stratospheric wind profiles should be used as bogus observations for the period from 1 July 1981 to 31 December 1998". In our revision we have added this reference here.

The information that the bogus data is used "at and above 30hPa" is from a personal communication and we have confirmed this by analyzing the datasets themselves (i.e. we found that there is almost no difference between ERA-40 and NCEP-CFSR values at and above 30 hPa). However, as Saha et al. (2010) did not explicitly mention the vertical range of the bogusing, we have made an appropriate modification to the sentence in our revision.

*(L10, P6) As the tropics are the focus for the paper, it would make sense to limit the latitudes in figure 3 to something like 20-30 degrees, for example. I do not think the extratropical SD differences show anything interesting anyway.*

We believe that showing the region outside the tropics provides a better context for the reader to see how exceptional the equatorial region is. In our revision we follow the reviewer's suggestion to the extent of reducing the latitude range shown in Fig. 3 from 90S-90N to 60S-60N.

*(L20, P6) Why have the authors chosen levels below 100hPa in figure 4? They should explain the reasoning behind this. Not much is written about this figure.*

As noted earlier we have removed the original Fig. 4 from the revised manuscript.

*(L25, P6) Perhaps refer here to 'Indonesia' or the 'maritime continent' rather than the ʹwarm pool'. Also, the central (through to western) Pacific is where the most conspicuous SD values reside. This clarification may be important in pointing to other sources of model disagreement: in particular modes of (ocean-)atmosphere variability (e.g. ElNino-Modoki).*

We followed this suggestion and in the revision we use "maritime continent" and "central Pacific" to denote these regions. .

*(L32, P7) Bogota data is on panel j not h. Also the authors should consider doing a F-test to compare the differences in variances with and without IGRA data.*

Thank you for noting this typo. We have conducted an F-test to see whether the SD of monthly

values computed over periods with IGRA data can be regarded as significantly different from the SD computed over periods without IGRA data. The results show that for several stations the differences between periods with and without IGRA data indeed can be regarded as significant. Specifically the SDs at Seychelles, Thiruvananthapuram, Ascension and Abidjan display homoscedasticity distributions, whereas the SDs at Nairobi, Menado, San Cristobal, Bogota, Manaus and Belem have heteroscedasticity distributions with and without IGRA data. In general, the stations with the largest (smallest) SD difference between periods with and without IGRA data have heteroscedasticity (homoscedasticity) distributions.

*(L21, P9) It is a pity that MERRA-2 is not shown here, whilst perhaps attempting to minimise the strong SAO signal (at 10hPa) afflicting a couple of years early in the time record. As that model has a spontaneous QBO it would be very interesting to see the timing of the E/W and W/E transitions. It would clearly be less reliant on analysis increments and sufficient observations to constrain the QBO phase progression.*

When we include MERRA-2 in the original Fig. 9 (and adjust for the fact that MERRA-2 only starts in 1980) then we get the Fig. R6 below, where the MERRA-2 results really look peculiar. We return to this issue in the discussion around Figure 18a (Fig. 16a of the revised version) that includes the later part of the MERRA-2 data record. This shows that the MERRA-2 reanalyses display an easterly-to-westerly phase transition at Singapore that is even more rapid than in the direct balloon observations. These results may indicate that the gravity wave sources in MERRA-2 are now excessive. We have extended our discussion here in the revised manuscript.

*(L8, P10) It is evident that all 5 reanalyses shown do actually show a positive anomaly near the maritime continent, although there is evident a local maximum in 3-4 of the datasets (probably highlighted due to contouring). Might also highlight (and reference) the fact a wave-1 warming occurred during December 1998 (and a wave-2 in February 1999)*

This is related with your comments above. Again, we have confirmed the result is not changed qualitatively when we analyze other years, in agreement with the earlier results of Hamilton et al. (2004).

*(L17, P10) subtitle: "Dependence of the ? Difference..."*

In our revised version we have changed this subtitle to "Difference depending on the QBO phase".

*(L29, P10) "The overall larger SD in the westerly QBO phase, as compared to the easterly QBO phase,..."*

In our revised version we followed your suggested change.

*(L7, P12) "...has reported stratospheric..."; reference, Pers. Comm.? The sentences following this need to be looked at. It is mentioned that a change in SD occurs around 1998, but then it is mentioned that there was a bias in the forward model of JRA-25. But presumably the bias in the forward model will not be responsible for changes around 1998 (forward model should not change during the reanalysis period – unlike operational analyses)*

Fujiwara et al. (2015) explained this in detail as "The radiative scheme used in the JRA-25 forecast model has a known cold bias in the stratosphere, and the TOVS SSU/MSU measurements do not

have a sufficient number of channels to correct the model's cold bias; after introducing the ATOVS AMSU-A measurements in 1998, such a cold bias disappeared in the JRA-25 data product". In our revised version we have added references to Fujiwara et al. (2015) and to the relevant work of Onogi et al. 2007.

*(L25..., p14) The authors should relax the statement referring to wavenumbers 1 and 2 dominating the 'eddy' zonal wind. Figure 10, mainly shows the zonal anomaly of zonal wind for January 1999 (a month sandwiched between two SSWs), so may not be representative of conditions at other times (e.g. 1979-2001)*

As noted earlier we have reason to regard this as describing typical circumstances. Again a number of individual cases are shown in the figures R3, R4 an R5 below.

**Reply to anonymous referee #4**

*General*
*This paper is part of a series of papers reporting on the comparison of the representation of different aspects in major global atmospheric reanalysis datasets; here the focus is on the tropical stratospheric zonal wind and the QBO. The paper works out in detail the agreement and disagreement between the reanalyses. It also analyses the impact of different observations on the quality of the reanalysis. I recommend accepting the paper for publication in ACP, after some critique (see below) has been taken into account in the revised version.*

Thank you for your positive evaluation.

*The paper states that most free-running GCMs have problems simulating a realistic QBO. While I agree that the QBO is still a challenge for GCMs, progress has been made in recent years. Models that have addressed the major shortcoming of GCMs in this respect, namely the representation of atmospheric waves that contribute to driving the QBO, in particularly waves with short vertical wavelengths have shown success in allowing a quasi-biennial periodicity to emerge. Of course parametrised wave drag is still required to generate a realistic QBO. I suggest discussing these recent developments (Orr et al., 2010; Anstey et al., 2016) in a bit more detail.*

In the revised manuscript we have added a short sentence to the Introduction explaining these recent developments.

*Further, the QBO induces a secondary meridional circulation, i.e. QBO variations in meridional and vertical winds (Punge et al., 2009). As this point is both important for tropical transport and intimately related to the QBO, I suggest extending the analysis in the paper somewhat to cover this aspect. I think this could be an important contribution of this paper.*

Abalos et al. (2015) have recently compared the stratospheric Brewer-Dobson circulation among MERRA, ERA-I and JRA55, while Miyazaki et al. (2015) did a somewhat similar study with 6 different reanalyses. Abalos et al. find quite large (~40%) differences among the reanalyses in the overall strength of the BD circulation and, specifically related to the QBO, they find that "There is a large spread among the estimates [of the fraction of variance in mean vertical motion explained by the QBO]". It is not clear that, even if we had the inclination to pursue this, we could add much to the Abalos et al. paper. In our revised manuscript we reference Abalos et al. and include a sentence on this issue in the summary section.

*On many instances the paper points out observations of differences and aspects of the reanalyses that are interesting to note. For example the finding that quasistationary waves differ significantly among reanalyses. However, it is more important to make progress on finding the reasons for differences between reanalyses. Any suggestions in the paper how to make progress in this direction would be very helpful. I believe that it is beyond the scope of the paper to do assimilation experiments removing certain data sets from the analysis and in this way exploring the impact of particular data sets on the "message" of the reanalysis. But such studies have been done. And perhaps this paper could at least suggest ideas how to move forward in this direction in a discussion. For example what could be important and relevant assimilation experiments to perform?*

While we certainly agree that the comparison of analyses produced with subsets of the total input data can be illuminating, we feel this is beyond the scope of our present paper which is strongly focused on what can be learned by intercomparing the monthly-mean zonal winds as represented in multiple state-of-the-art global reanalysis data sets. Differences between individual reanalyses are not just related to which observations were actually used, but might also depend on how a given data set was used, e.g., what data-quality-control and bias-correction procedures were actually applied or not applied. These procedures are different in the details for different reanalyses, and we cannot obtain the detailed information about this (see our response on this issue to referee #3 above).

*In summary, I think the paper could be improved with respect to some aspects in the revision. I think it will be a valuable contribution to ACP.*

Thank you for your positive evaluation.

*Minor issues*
*· p. 3, l 6: 'most such models'?*

In the revision we changed this to the explicit "most GCMs".

*· p. 4, l. 5: there are also more recent publications on this point*

In the revised version we have added a reference to Abalos et al. 2015 and Miyazaki et al. 2015.

*· p 7., l 17: So is this only an expectation?*

In the revision we removed "expected" here, which makes our meaning clearer.

*· p 7., l 32: off?*

We meant "off the equator".

*· p 9, l 4: if one writes $-$ in LATEX, then one obtains proper minus signs*

We actually are using MS-Word for our manuscript, and we will be careful to make sure this looks OK in the final version.

*· p 10, l 7: change 'represents' to 'shows'*

In our revision we followed this suggestion.

*· p 10, l. 26: continuous over which time period?*

We mean continuous over the 1979-2001 period under discussion here (e.g. Fig. 6b).

*· p 11, l. 12: I agree it is interesting, but what is the conclusion here?*

We may be seeing the effects of the low numbers of in situ observations in both the upper troposphere and stratosphere in the central Pacific region.

*· p 11, l. 20: I agree it is interesting, but what is the conclusion here?*

As we mentioned in conclusion section, our study has provided detailed results showing that the high accuracy and high resolution wind measurements by in situ radiosondes provide very important constraints in the reanalyses of circulation in the tropical stratosphere.

*· p 11, l. 23: why is there a drop?*

As we mention a few sentences later, this drop is reasonably attributed to improved satellite observations (AMSU-U).

*· p 12, l 5: How likely is the possible reason?*

We cannot be definitive, of course, but the number of radiosonde observations did not dramatically increase around 1998 as shown in Fig. 15 (Fig. 17 in the original manuscript), thus implicating the upgraded satellite observations as the likely main cause..

*· p 12, l 7: change 'that' to 'on'*

We have made this change in the revised version.

*· p 12, l 8, 9: I think you mean the radiation code of the forecast model here.*

Yes the reviewer is correct. In our revision we have modified this sentence.

*· p 12, l 29: Discuss how this could be tested.*

As you indicated, doing assimilation experiments removing certain data sets from the analysis and exploring the impact of particular data sets is one possible way forward. Comparison between JRA-55 and JRA-55C might be possible, although it is beyond the scope of the present study and remains in the future. We add this sentence in the summary and concluding remarks section.

*· p 13, l 14: the three most modern reanalyses*

In our revision we have made this modification.

*· p 14, l 4: why only 'nearly'?*

We have confirmed the zonal wind in NCEP-CFSR is very similar to that in ERA-40, but not perfectly identical. The ERA-40 analyses were used as bogus data above 30 hPa in the NCEP-CFSR reanalyses.

*· p 14, l 12: why?*

As shown in Fig 11, the SD has zonally more non-uniform structure in the westerly phase of the QBO at 10 hPa, but the mean SD becomes more zonally uniform (Fig. 4 or original Fig. 5). The penetration of the Walker circulation, which is not related with the QBO phase, does not appear to influence the circulation in the 10-30 hPa range (Fig. 13 in the original manuscript).

*· p 14, l 17, 18: It also means that the constraint is really necessary. So there are problems with the underlying model, correct?*

Yes, that is our interpretation as well.

*· p. 16, Durre et al: abbreviate first names Fig 2, bottom panel: lines are difficult to disentangle*

In the revision we have made this suggested modification.

**Reply to short comment by Dr. A. Hertzog**

*In this detailed study, Kawatani et al. (2016, hereinafter referred to as K16) present an intercomparison of monthly-mean zonal winds in the tropics in recent reanalyses. While focusing on monthly means is fully justified when tropical large-scale circulations (like the Quasi-Biennial Oscillation or quasi-stationnary planetary waves) are addressed, I would like to emphazise that such a time average ignores a significant fraction of the wind variability in the tropical lower stratosphere. Indeed, propagating disturbances associated with planetary waves trapped in the equatorial wave guide (e.g., Kelvin and Rossby-gravity waves) are essentially discounted in K16, even though reanalyses in principle have sufficient horizontal and temporal resolution to resolve most of these waves. It might thus be appropriate that K16 briefly discuss the implications of such time averaging on their results. Two references are provided hereinunder to that purpose, and also to bring up some elements on reanalysis agreement vs accuracy.*

*1. K16 show that the agreement between reanalyses in monthly-mean zonal winds has continuously improved since 1979. The standard deviation (SD) among reanalyses reaches zonal-mean values of ~1 ms$^{-1}$ at 70 hPa in the last decade they studied (2001-2011), and never exceeds 1.8 ms$^{-1}$ locally (their Figures 14 and 15). Despite the difficulties of constraining the tropical-stratosphere dynamics in atmospheric models with observations (which are recalled by K16), these SDs are quite impressive, as they are less than the assumed uncertainty associated with radiosonde winds in most models during the assimilation process. Yet, such good agreement likely does not apply to instantaneous reanalyses (i.e., without monthly average): Baker et al. (2014) (their Figure 2) have for instance shown that, in 2010, the zonal-mean SD between ECMWF operational analyses and NCEP GFS in zonal winds at 300 hPa is typically 3 ms$^{-1}$, and can reach values over 5 ms$^{-1}$ over the eastern Pacific and Indian Oceans, i.e. at least three times the values reported in K16: according to K16, the SD at 300 hPa should be less than in the lower stratosphere (their Figure 1 and 13).*

*2. Away from regions with assimilated observations, an agreement between reanalyses does not necessarily mean an equivalent agreement with observations, even in the most recent decade. For instance, Podglajen et al. (2014, hereinafter referred to as P14), who compared reanalyzed winds with independent in-situ observations performed along long-duration balloon flights in 2010, have reported occurrences where ECMWF operational analysis, ERA-interim and MERRA products all agree, while the balloon observations depart from them (see their Figure 4). These events, which are associated with equatorial waves, induce discrepancies between reanalyses and observations that can reach values as large as 10 ms$^{-1}$ and last for weeks. They once again tend to occur over areas with (very) few radiosounding stations: the Eastern Pacific and Indian Ocean. In these areas, observational increments are very low (about 1 ms$^{-1}$ or less, see Figure 11 in P14), and the model dynamics is essentially running freely in the lower stratosphere.*

*It may therefore be worthwhile to warn the readers that they should not over-interpret encouraging figures regarding the improved agreement between reanalyses reported in K16.*

In the revised manuscript, we explicitly caution against over-interpreting our results which are based solely on monthly mean fields. In the summary section, we explain as follows:
"The difference among reanalyses using twice daily data should be much larger than our SD based on monthly mean data (cf. Baker et al. 2014). Padglajen et al. (2014) compare reanalyses winds with independent *in situ* observations performed along long-duration balloon flights. They report that ERA-I and MERRA represent similar disturbances associated with equatorial waves, but reanalyses depart from the balloon observations. As the present study focus on monthly mean field, our analyses ignore variability with shorter time scales".

*I finally note that the lower agreement between reanalyses during QBO shear phases reported in K16 likely has a counterpart in the agreement between reanalyses and observations, as discussed in P14. Shear layers indeed tend to reduce the vertical wavelengths of waves that propagate in the shear direction while they increase the associated horizontal-wind disturbances. The reduced wavelength means that the model resolution may become insufficient to properly resolve the wave disturbances, even though the wave signal is present in the assimilated observations (see for instance Figure 9 in P14).*

Yes we agree that the effect of mean wind variations on the wavelength of vertically-propagating waves can be a contributor to the dependence we found of SD on the QBO phase.

**Six figures we prepared for this reply to reviewers**

[Figure]

Fig.R1: Latitude-height cross-sections of the standard deviation among (a) four reanalyses of ERA-I, JRA-25, JRA-55, and MERRA, (b) three analyses of ERA-I, JRA-55, and MERRA and (c) three analyses of ERA-I, JRA-55, and MERRA-2 from 1980 to 2012.

[Figure]

Fig. R2: Longitude-height cross section of the correlation between absolute value of zonal wind and the standard deviation among reanalyses in 10ºN-10ºS. Contour interval is 0.1 and correlations with 95% significance are shaded.

[Figure]

Fig. R3: Longitudinal variations of the *u'* (deviation of the zonal wind values from the zonal mean) at 10hPa over the equator in January from 1990 to 2000. The zonal wind values are shown on the top right corner. W and E mean westerly and easterly, respectively.

[Figure]

Fig. R4: Deviation of the zonal wind values from the zonal mean (*u'*) at 10hPa of five reanalyses in January (left) 1992 and (right) 1993 when the five reanalyses averaged equatorial zonal mean zonal wind is westerly over the equator.

[Figure]

Fig. R5: Deviation of the zonal wind values from the zonal mean (*u'*) at 10hPa of five reanalyses in January (left) 1995 and (right) 1997 when the five reanalyses averaged equatorial zonal mean zonal wind is westerly over the equator.

[Figure]

Fig.R6: The same as Figs.9a-d in the original manuscript but including MERRA-2 for 1979-2001. Due to overestimation of the SAO, 10hPa MERRA-2 zonal wind has large errors and it becomes difficult to discuss possible bias resulting from QBO phase transitions.

---

## Author Comment (AC2) · 10 May 2016

We thank the referee for the helpful comments. Please find attached our reply.

Please also note the supplement to this comment:
http://www.atmos-chem-phys-discuss.net/acp-2016-76/acp-2016-76-AC2-supplement.pdf

---

## Author Comment (AC6) · 10 May 2016

I am sorry that the title was wrong. This comment is for Dr. Hertzog.

---

## Author Response (AR2)

**Authors comments on "Representation of the Tropical Stratospheric Zonal Wind in Global Atmospheric Reanalyses" by Y. Kawatani, K. Hamilton, K. Miyazaki, M. Fujiwara, J. Anstey**

Corresponding author: Yoshio Kawatani (yoskawatani@jamstec.go.jp)

We are pleased to hear that our manuscript is accepted for publication in ACP after technical corrections. We have revised the manuscript following Prof. Haynes's suggestions. In this reply, we write the *editor's comments in blue italics*, while our responses are in regular fonts.

*p4 l25: 'Furthermore, it is not feasible to determine exactly what observational data were actually assimilated at each data assimilation analysis step (e.g., what data quality control and bias correction procedures were actually applied).' One of the referees criticised this statement -- taking the view that such information must be available somewhere. One issue is availability of information. Of course, each re-analysis was introduced by some kind of publication and you have cited those publications at the beginning of Section 2. But I guess that these publications do not contain absolutely every detail of the reanalysis. A distinct issue is any publication can only have finite scope -- not every detail can be covered. Which are you trying to emphasise? Perhaps you are trying to emphasise both? A minor change of text to something like -- 'However, we believe it is interesting to investigate representations of key phenomena in the reanalyses and we hope that such investigation ...' might give a reader a more positive impression.*

We change the text following your suggestion as follows at p4l25.
"However, we believe it is interesting to investigate representations of key phenomena in the reanalyses and we hope that such investigation will contribute to basic understanding and to improving future reanalysis products"

*p6 l19: Another topic raised by referees was re MERRA-2. You justify why you don't give MERRA-2 full consideration, but that could be a bit clearer. For example, before you say 'For these reasons, our study focuses mainly on the five reanalyses: ERA-40, ERA-I, JRA-25, JRA-55 and MERRA.' — it perhaps confusing to mention immediately before that MERRA-2 has improved the QBO. I suggest that you make it clear that the reasons are provided in the previous 2 paragraphs (not just the same paragraph) and that you move sentence about improved QBO to after the statement above. -- e.g '(But note that MERRA-2 shows some improvement ....)'*

We move sentence about improved QBO and change the text following your suggestion as follows at p6l19.

"For these reasons, our study focuses mainly on the five reanalyses: ERA-40, ERA-I, JRA-25, JRA-55 and MERRA. But note that the MERRA-2 zonal winds show improved representation of the QBO compared to MERRA at 30–50 hPa (Lawrence Coy et al., 2016)"

*p17 l25: 'Padglajen' > 'Podglajen' (as in reference list)*

We have made this change in the revised version.

We also change the reference of *Miyazaki et al. (2015) ACPD* to *Miyazaki et al. (2016) ACP*, which is just published on 20 May 2016.

[revised manuscript text omitted]